# Sliding Puzzles Gym: A Scalable Benchmark for State Representation in Visual Reinforcement Learning

**Bryan L. M. de Oliveira** [1 2]   **Luana G. B. Martins** [1]   **Bruno Brandão** [1 2]   **Murilo L. da Luz** [1 2]
**Telma W. de L. Soares** [1 2]   **Luckeciano C. Melo** [1 3]

## Abstract

Effective visual representation learning is crucial for reinforcement learning (RL) agents to extract task-relevant information from raw sensory inputs and generalize across diverse environments. However, existing RL benchmarks lack the ability to systematically evaluate representation learning capabilities in isolation from other learning challenges. To address this gap, we introduce the Sliding Puzzles Gym (SPGym), a novel benchmark that transforms the classic 8-tile puzzle into a visual RL task with images drawn from arbitrarily large datasets. SPGym's key innovation lies in its ability to precisely control representation learning complexity through adjustable grid sizes and image pools, while maintaining fixed environment dynamics, observation, and action spaces. This design enables researchers to isolate and scale the visual representation challenge independently of other learning components. Through extensive experiments with model-free and model-based RL algorithms, we uncover fundamental limitations in current methods' ability to handle visual diversity. As we increase the pool of possible images, all algorithms exhibit in- and out-of-distribution performance degradation, with sophisticated representation learning techniques often underperforming simpler approaches like data augmentation. These findings highlight critical gaps in visual representation learning for RL and establish SPGym as a valuable tool for driving progress in robust, generalizable decision-making systems.

## 1. Introduction

Learning meaningful representations from raw sensory inputs, such as visual data, is fundamental to reinforcement learning (RL) agents' ability to generalize across different tasks in complex, open-world environments (Bengio et al., 2013; Lesort et al., 2018). In visual RL, agents must process high-dimensional pixel data, extract useful features, and utilize these features for decision-making (Mnih et al., 2015; Yarats et al., 2021a). This becomes especially crucial as real-world applications demand adaptability to unstructured and diverse observations. However, measuring an agent's representation learning capabilities independently of other learning tasks, such as policy optimization or dynamics modeling, remains a key challenge in RL benchmarks.

While traditional RL benchmarks such as Atari (Bellemare et al., 2013) and DeepMind Control Suite (Tassa et al., 2018) offer valuable evaluation platforms for overall agent effectiveness, their performance metrics inherently intertwine representation learning with policy optimization and environment dynamics. More recent dedicated benchmarks for visual learning, though promising, fall short in their capacity to precisely modulate visual complexity. For instance, ProcGen (Cobbe et al., 2020) simultaneously modifies both visual and task difficulty, obscuring the specific impact of representation learning, while the Distracting Control Suite (Stone et al., 2021) introduces visual distractors that are unrelated to the main task and can be safely disregarded by the model. Consequently, current benchmarks lack the necessary precision to systematically assess an agent's ability to acquire task-relevant visual representations.

To address this gap, we introduce the *Sliding Puzzles Gym (SPGym)*[1], an open-source benchmark created to evaluate how agents handle increasingly diverse visual observations across training runs. As depicted in Figure 1, SPGym transforms the classic 8-tile puzzle into a visual RL challenge through three central design principles: (1) preserving consistent environment dynamics regardless of difficulty, ensuring that the underlying task remains unchanged; (2) enabling

---

[1]Advanced Knowledge Center for Immersive Technologies – AKCIT, Brazil [2]Institute of Informatics, Federal University of Goiás, Goiânia, Brazil [3]OATML, University of Oxford, United Kingdom. Correspondence to: Bryan L. M. de Oliveira <bryanlincoln@discente.ufg.br>.

*Proceedings of the 42$^{nd}$ International Conference on Machine Learning*, Vancouver, Canada. PMLR 267, 2025. Copyright 2025 by the author(s).

---

[1]Available at https://github.com/bryanoliveira/sliding-puzzles-gym.

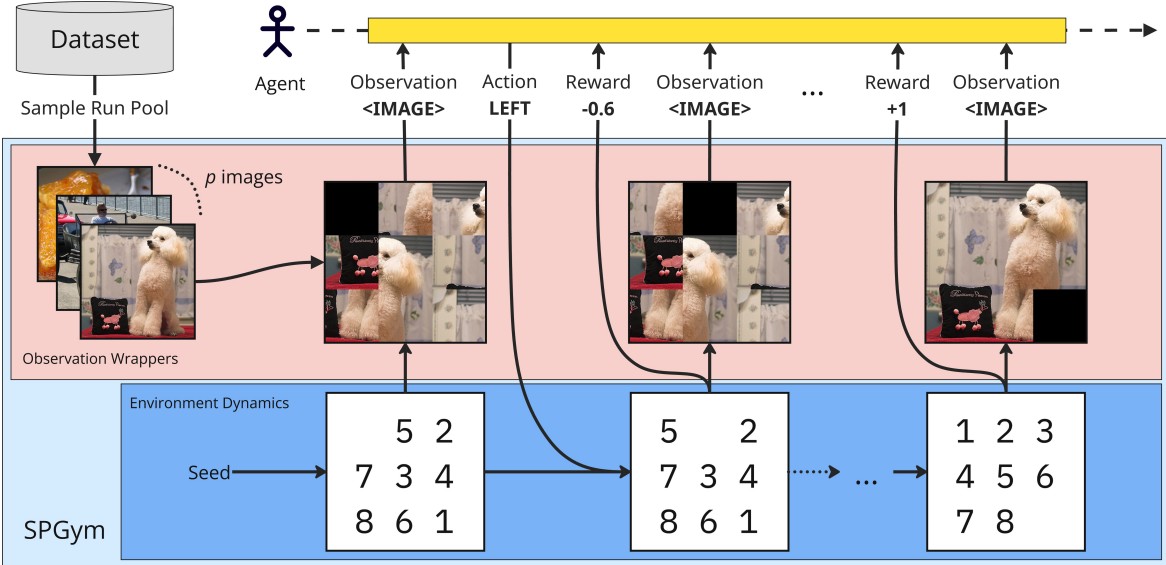

*Figure 1.* **Overview of SPGym.** The framework extends the 8-tile puzzle by replacing numbered tiles with image patches. At each training run, SPGym samples a pool of images and, at each episode, it randomly selects one of those images to form the observations. While we scale visual diversity by adjusting the pool size, the task and environment dynamics remain fixed.

precise control over visual complexity through adjustable grid dimensions and image pool sizes; and (3) establishing a clear success metric based on puzzle completion. This carefully engineered framework provides researchers with a controlled environment to systematically examine how agent performance degrades as visual diversity increases, offering insights into the fundamental limitations of their representational abilities while learning from experience.

Our experiments show that SPGym effectively distinguishes agents by their out-of-the-box representation learning abilities, with strong correlations between linear probing accuracy of learned encoders and task performance. While pretraining and data augmentation are beneficial, many advanced methods with standard configurations for discrete visual RL underperform, suggesting either a need for task-specific tuning or fundamental mismatches with SPGym's visual-structural dynamics. More concerningly, our tiered generalization analysis exposes fundamental limitations: on SPGym, agents that master the training images fail to transfer to unseen ones, even when trained on larger and more diverse image pools. In fact, performance often degrades as visual diversity increases, indicating that current methods struggle to learn truly generalizable representations. We further demonstrate SPGym's extensibility through experiments with procedurally generated images and larger puzzles, maintaining fixed observation/action spaces while expanding state space and visual diversity. These findings expose critical gaps in current visual RL methods and establish SPGym as a valuable tool for advancing robust, generalizable artificial agents.

**Contributions.** We make three key contributions to visual RL research: (1) we introduce SPGym, a novel benchmark that systematically evaluates representation learning by scaling visual complexity while keeping environment dynamics constant; (2) we conduct an extensive empirical analysis of state-of-the-art methods, uncovering critical limitations in their ability to process diverse visual inputs; and (3) we provide fundamental insights into the challenges of scaling visual RL, offering directions for advancing representation learning in decision-making systems.

## 2. Related Work

**Traditional RL benchmarks.** Visual RL environments such as the Arcade Learning Environment (Bellemare et al., 2013), DeepMind Control Suite (Tassa et al., 2018), DeepMind Lab (Beattie et al., 2016), and CARLA (Dosovitskiy et al., 2017) are widely used for pixel-based agent evaluation. These platforms have driven progress in visual representation learning for decision-making, including pretraining (Higgins et al., 2017; Stooke et al., 2021; Schwarzer et al., 2021), contrastive methods (Laskin et al., 2020b), self-supervised prediction (Schwarzer et al., 2020), and world models (Hafner et al., 2025). However, by measuring overall agent effectiveness, which relies on learning representations, policies, and dynamics, the specific impact of advances in representation learning for RL may be obscured. SPGym addresses this drawback by regulating visual complexity while maintaining constant environment dynamics, allowing for targeted assessment of representation learning approaches for decision-making.

**Specialized benchmarks for visual RL.** Recent benchmarks have sought to better assess visual generalization and robustness of RL agents. ProcGen (Cobbe et al., 2020) employs procedurally generated levels, but its difficulty scaling simultaneously affects both visual and task complexity. The Distracting Control Suite (Stone et al., 2021) evaluates robustness to visual variations, but these serve as task-irrelevant distractions that agents can safely learn to ignore while solving core tasks. COOM (Tomilin et al., 2023) provides a continual learning benchmark for embodied pixel-based RL in 3D environments, focusing on catastrophic forgetting and knowledge transfer across tasks rather than isolating representation learning in a single task. However, these approaches do not make visual understanding essential for task success. SPGym overcomes this limitation by making visual coherency fundamental to puzzle solving while maintaining fixed task complexity. Through controlled scaling of visual diversity in puzzle tiles, SPGym ensures that the representation learning challenges are tied to task completion, enabling a more precise evaluation of RL agents' visual learning capabilities.

**Puzzle-based benchmarks.** Estermann et al. (2024) demonstrated puzzle-based environments' value for evaluating neural algorithmic reasoning, focusing on discrete state spaces. Though they tested pixel observations with sparse rewards, agents struggled with basic visual inputs. This work revealed puzzles' potential and challenges for visual learning, showing the need for systematic representation learning evaluation in these controlled settings. SPGym advances this research direction by incorporating rich visual observations while maintaining the controlled nature of puzzle environments, facilitating systematic assessment of representation learning in the visual domain.

**Methods for solving the sliding tile puzzle.** Classical approaches to sliding puzzles employ search algorithms such as A* and IDA*, leveraging domain-specific heuristics like the Manhattan distance (Korf, 1985; Burns et al., 2012; Lee & See, 2022). These methods guarantee optimal solutions but require direct access to the internal puzzle state and are often computationally intensive. Deep RL offers a scalable alternative by learning effective strategies without hand-crafted heuristics (Agostinelli et al., 2019; Moon & Cho, 2024; Estermann et al., 2024), though prior work has primarily focused on discrete state representations rather than learning directly from visual observations (Agostinelli et al., 2019; Moon & Cho, 2024). We build on these methods by using the Manhattan distance as the basis for our reward function, providing a well-shaped learning signal for RL agents. In SPGym, we evaluate standard RL algorithms where agents must learn solely from pixel observations, without access to internal states, to assess their out-of-the-box performance in this challenging setting.

## 3. The Sliding Puzzles Gym

SPGym extends the classic sliding tile puzzle, a game where players rearrange shuffled tiles on a grid by sliding a tile into an adjacent empty space, with the goal of restoring an ordered configuration (Figure 1). Our framework generalizes this setup by supporting configurable $H \times W$ grid dimensions (from $2 \times 2$ upwards) and diverse observation modalities. In our experiments, we primarily use a $3 \times 3$ grid where tiles are image patches, and the agent receives the composite image of the grid as input. Our implementation builds on the Gym (Brockman, 2016) interface for modularity between environment and agent.

**Formalization.** We formulate SPGym as a partially observable Markov decision process (POMDP) defined by the tuple $(\mathcal{S}, \mathcal{A}, \mathcal{P}, \mathcal{R}, \mathcal{S}_0, \Omega, \mathcal{O})$, where $\mathcal{S}$ is the finite state space representing all solvable puzzle configurations, $\mathcal{A}$ is the discrete action space (tile movements), $\mathcal{P} : \mathcal{S} \times \mathcal{A} \to \mathcal{S}$ defines the deterministic transition dynamics, $\mathcal{R} : \mathcal{S} \to \mathbb{R}$ is the reward function, $\mathcal{S}_0$ is the initial state distribution (e.g., uniform over $\mathcal{S}$), $\Omega$ is the observation space, and $\mathcal{O} : \mathcal{S} \times \mathcal{I} \to \Omega$ is the observation function parameterized by a data pool $\mathcal{I}$, which in this work consists of images.

**State space and observations.** Crucially, agents do not have direct access to states $s \in \mathcal{S}$ and must learn policies based solely on observations in $\Omega$. For visual observations, each training run begins by sampling $p$ images from a predefined dataset to form an image pool $\mathcal{I}$. At the start of each episode, we select a random image $i \in \mathcal{I}$, partition it into $H \times W$ distinct patches corresponding to the puzzle tiles, and assign each patch to a tile position according to the current state $s$. The observation function $\mathcal{O}(s, i)$ then renders the composite image by arranging these patches according to the puzzle configuration. The agent's task is to reconstruct the original image, testing its ability to form compositional representations from visual input.

This formulation provides two independent mechanisms for controlling complexity: (1) varying the pool size $p$ adjusts the diversity of observations by changing the number of available images in $\mathcal{I}$, directly affecting $|\Omega|$, and (2) modifying grid dimensions $H \times W$ alters both the state space size $|\mathcal{S}|$ and the visual complexity of individual observations. Both mechanisms operate while keeping the underlying transition dynamics $\mathcal{P}$, action space $\mathcal{A}$, and reward function $\mathcal{R}$ invariant. SPGym is dataset-agnostic, supporting any image dataset, including procedurally generated ones.

**Action space and dynamics.** The action space $\mathcal{A}$ contains four possible actions: *UP*, *DOWN*, *LEFT*, or *RIGHT*, which move the corresponding tile to the empty space in a discrete manner. Once the agent selects a tile to move, the dynamics $\mathcal{P}$ define the next puzzle state in a predictable and

deterministic way, as illustrated in Figure 1.

**Reward function.** The reward function $\mathcal{R}$ provides dense feedback to guide agent learning while accommodating the non-monotonic nature of optimal puzzle-solving paths. Following established approaches in the sliding puzzle literature (Korf, 1985; Burns et al., 2012; Lee & See, 2022; Moon & Cho, 2024), we base our reward on the Manhattan distance between each tile's current position and its target position, normalized across all tiles. This distance metric is widely adopted in computational solutions to sliding puzzles due to its efficiency and effectiveness in capturing progress toward the goal state. Specifically, at each time step, the reward is computed as follows:

$$\mathcal{R}(s) = \begin{cases} -D, & \text{if action is valid} \\ -1, & \text{if action is invalid} \\ +1, & \text{if puzzle is solved} \end{cases} \text{, with} \quad (1)$$

$$D = \frac{\sum_{i=1}^{H}\sum_{j=1}^{W} |u_{i,j} - u_{i,j}^*| + |v_{i,j} - v_{i,j}^*|}{\sum_{i=1}^{H}\sum_{j=1}^{W} \max(i, H-i) + \max(j, W-j)}. \quad (2)$$

Here, $(u_{i,j}, v_{i,j})$ represents the current position of the tile at index $(i,j)$, $(u_{i,j}^*, v_{i,j}^*)$ is its target position. $D$ is the normalized Manhattan distance between current and target positions. Invalid actions, such as attempting to move a tile outside the grid boundaries, result in a penalty of $-1$ and don't alter the puzzle state. For example, in Figure 1, the *DOWN* and *RIGHT* actions would be invalid for the first state. Successfully solving the puzzle rewards the agent with $+1$ and terminates the episode. This formulation provides a well-behaved learning signal between $[-1, +1]$ and encourages solving the puzzle in minimal steps.

**Initial state distribution.** SPGym's initial state distribution $\mathcal{S}_0$ can be defined through two methods. The primary method generates a uniformly random $H \times W$ array where all tiles (including the blank tile) are placed randomly, and ensures solvability by adjusting the puzzle's parity through swapping the first two tiles when needed (Johnson & Story, 1879). The second method, designed to support curriculum learning, starts from a solved state and applies a sequence of random valid moves to reach the initial configuration. Although this approach guarantees solvability due to the reversibility of moves, it becomes computationally expensive for larger grid sizes due to the sequential nature of move application. Given these efficiency considerations, we exclusively employ the first method in our experiments.

**Diversity scalability and extensibility.** As previously mentioned, SPGym provides two orthogonal mechanisms for scaling task difficulty: visual diversity and grid size.

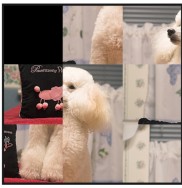

| State | One-hot | Image Overlay |

*Figure 2.* **Different observation modalities in SPGym.** Each modality presents a unique challenge for representation learning. The three presented observations represent the same puzzle state. We focus our experiments on image-based observations.

While our primary experiments use $3 \times 3$ image-based puzzles, SPGym supports larger grids and other observation modalities, such as one-hot encodings (Figure 2), to isolate specific research questions.

The primary scaling mechanism in SPGym isolates the challenge of representation learning by increasing visual diversity, illustrated in Figure 3, while keeping all other task components fixed across training runs. As the image pool size grows, agents require more samples to solve the task (Table 1), even though the state space, action space, and transition dynamics remain unchanged. By design, this increased difficulty stems directly from the representation learning challenge. We further confirm this link through linear probe analysis (Appendix C.1.2), which shows a strong correlation between representation quality and task performance. This controlled scaling provides a stress test for visual learning that is not possible in settings with fixed visual inputs or state-based observations (see Appendix C.1.1).

The secondary scaling mechanism modulates the search challenge by adjusting grid dimensions. Increasing the grid size from $3 \times 3$ to $4 \times 4$ dramatically expands the state space complexity and, as a result, the number of steps required to solve the task (see Table 3). For example, $3 \times 3$ puzzles have approximately $\frac{3^2!}{2} = 1.81 \times 10^5$ possible states, whereas $4 \times 4$ puzzles pose a much greater challenge with roughly $\frac{4^2!}{2} = 10^{13}$ possible states. This increase in grid size also makes representation learning more difficult, since the agent must correctly map and position a greater number of image patches, modeling a larger observation space. However, the size and shape of each rendered observation (i.e., the observation dimensions) remain unchanged, as does the action space (still four directional moves), and the task dynamics continue to be deterministic and fully predictable. While this scaling mechanism enables evaluation of exploration and policy-learning capabilities, our experiments show that $3 \times 3$ grids already provide sufficient discriminative power for assessing visual representation learning.

By maintaining consistent environment dynamics across these scaling mechanisms, SPGym creates a controlled ex-

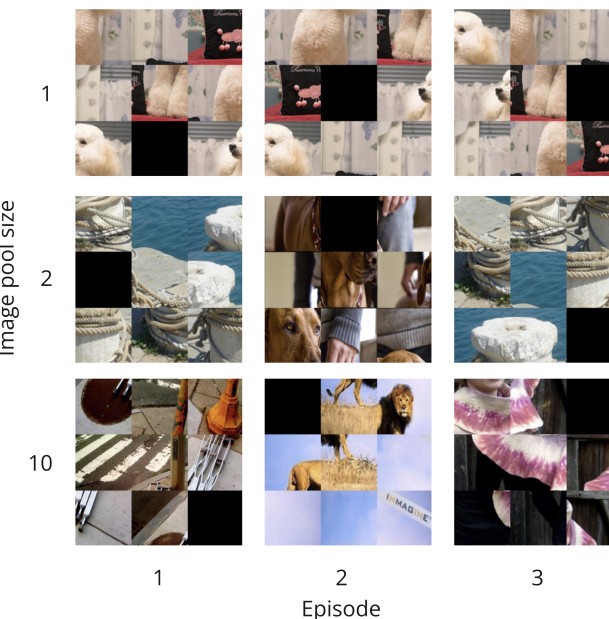

*Figure 3.* **The visual diversity scalability of SPGym.** Each row presents the first observation of 3 different episodes of the same training run. The visual diversity scales with the size of the image pool. Crucially, we keep the grid size fixed, ensuring that the increased difficulty comes solely from the agent's need to handle a larger variety of visual observations.

perimental setting where performance variations can be more directly attributed to an agent's representation learning and generalization capabilities.

## 4. Methods

In this section, we detail the evaluation protocol for various RL agents within SPGym, focusing on their ability to handle increasing visual diversity across training runs.

**Experimental setup.** We sample images from ImageNet-1k's validation split (Russakovsky et al., 2015) to construct visual observations, resizing each image to $84 \times 84$ pixels and normalizing values to $[0, 1]$. To isolate visual diversity, we fix the puzzle size to $3 \times 3$ and vary only the image pool size $p$. For each training run, we randomly sample $p$ distinct images to create a fixed image pool $\mathcal{I}$. At episode start, we randomly select one image from this pool to generate the puzzle observations. We cap the number of environment steps to 10M and limit episodes to 1,000 steps.

Our analysis focuses on sample efficiency, measured by the number of environment steps required to solve the puzzle (lower is better). We calculate this metric by averaging the steps needed to reach 80% success rate across all parallel environments in a run, then average this number across seeds. We terminate training runs early when an agent maintains

100% success rate for 100 consecutive episodes, indicating task completion. This early termination serves two purposes: it enables out-of-distribution evaluation before extreme encoder overfitting occurs, and it reduces computational costs for running the comprehensive set of experiments. Each experiment is conducted 5 times with different random seeds, and we report the mean ± 1.96 standard errors (95% confidence interval).

We evaluate sample efficiency using pools of 1, 5, and 10 images across all agent variants. For standard agents, we additionally test with progressively larger pools up to 100 images or until final performance is less than 80% success rate after the full 10M training steps. This protocol allows us to analyze both the effectiveness and scalability of different algorithms and representation learning methods.

**Algorithms and variants.** We explore three distinct algorithmic approaches: Soft Actor-Critic (SAC) (Haarnoja et al., 2018a), Proximal Policy Optimization (PPO) (Schulman et al., 2017), and DreamerV3 (Hafner et al., 2025), each representing different strategies for learning from visual observations. For each algorithm, we evaluate several representation learning variants to assess their effectiveness in handling visual diversity.

For SAC, we begin with the standard implementation for discrete action spaces proposed by Christodoulou (2019). We then examine several representation learning variants: RAD (Laskin et al., 2020a), which employs data augmentation; CURL (Laskin et al., 2020b), which uses contrastive learning; SPR (Schwarzer et al., 2020), which incorporates self-supervised prediction; DBC (Zhang et al., 2021), which focuses on state metric learning; SAC-AE and SAC-VAE (Yarats et al., 2021b), which utilize reconstruction-based learning; and a simple baseline (SB) proposed by Tomar et al. (2023), which implements a simplified approach with reward and transition prediction.

For PPO, we evaluate three encoder configurations: the standard version with random initialization, a variant pretrained on the same image distribution (in-distribution, ID), and a variant pretrained on a different image distribution (out-of-distribution, OOD). These pretrained variants respectively provide an upper bound on expected pretraining performance and help establish the potential benefits of generally pretrained encoders, though we acknowledge this may make direct sample efficiency comparisons with other methods less fair.

For DreamerV3, we compare the standard version against a variant without decoder gradients to evaluate the impact of the reconstruction objective on performance. Detailed descriptions for all agents and their learning objectives are provided in Appendix B.

**Hyperparameters.** RAD, CURL, and SPR require data augmentation. We apply these augmentations to observations after sampling them from the replay buffer. For each algorithm, we conducted individual augmentation searches with the objective of maximizing sample efficiency (detailed in Appendix A.4.2). These experiments consistently converged to a simple two-step pipeline, which we use throughout all evaluations: first converting to grayscale with 20% probability, then randomly shuffling the color channels.

We evaluate out-of-the-box performance of existing approaches in SPGym by adopting neural architectures and hyperparameters from established visual discrete control implementations. Our base architecture uses three-layer CNN encoders with mirrored deconvolutional decoders, while actor, critic, and auxiliary components employ multi-layer perceptrons (MLPs).

For SAC-based agents, we follow Yarats et al. (2021b) and Tomar et al. (2023) by blocking actor gradients through the encoder to prevent representation collapse, while allowing critic and auxiliary gradients. PPO and DreamerV3 maintain their original gradient flow patterns. While PPO and DreamerV3 worked robustly with default configurations, SAC-based agents required tuning of the temperature parameter $\alpha$ (Appendix A.4.1). We found a fixed value of 0.05 to work best across all SAC variants, as the automatic tuning from Haarnoja et al. (2018b) proved ineffective here.

Where applicable, we preserve uniform hyperparameters while drawing algorithm-specific configurations from their respective source papers. The publicly available code repository[2] extends both CleanRL (Huang et al., 2022) and the official DreamerV3 codebase. Additional methodological details are provided in Appendix A.

## 5. Results

Our analysis reveals three fundamental tensions in visual RL: between method assumptions and environment structure, sample efficiency and solution optimality, and training diversity versus generalization capability. We organize findings through seven research questions.

### 5.1. Can SPGym distinguish agents on their approaches to representation learning?

Table 1 demonstrates SPGym's ability to differentiate agents based on their representation learning approaches across varying levels of visual diversity. Our primary goal in this analysis is to compare the sample efficiency with which different agents learn useful visual representations.

To explore the potential benefits of leveraging pretrained

---

[2]Accessible at https://github.com/bryanoliveira/spgym-experiments

Table 1. **Million steps to reach 80% success rate across pool sizes during training.** Lower is better. Best performing variant for each algorithm and pool size is highlighted in bold.

| Agent | Pool 1 | Pool 5 | Pool 10 |
|---|---|---|---|
| PPO | $1.75_{\pm0.44}$ | $7.80_{\pm1.08}$ | $9.73_{\pm0.36}$ |
| PPO + PT (ID) | $\mathbf{0.95_{\pm0.21}}$ | $\mathbf{5.55_{\pm1.22}}$ | $\mathbf{9.17_{\pm1.10}}$ |
| PPO + PT (OOD) | $1.34_{\pm0.42}$ | $7.03_{\pm1.07}$ | $9.70_{\pm0.41}$ |
| SAC | $0.33_{\pm0.07}$ | $0.91_{\pm0.12}$ | $2.03_{\pm0.38}$ |
| SAC + RAD | $\mathbf{0.24_{\pm0.03}}$ | $\mathbf{0.42_{\pm0.06}}$ | $\mathbf{0.82_{\pm0.18}}$ |
| SAC + CURL | $0.46_{\pm0.10}$ | $1.56_{\pm0.31}$ | $5.24_{\pm1.92}$ |
| SAC + SPR | $2.09_{\pm0.81}$ | $3.68_{\pm1.68}$ | $10.00_{\pm0.00}$ |
| SAC + DBC | $0.99_{\pm0.25}$ | $1.12_{\pm0.22}$ | $2.13_{\pm0.41}$ |
| SAC + AE | $1.04_{\pm0.24}$ | $1.02_{\pm0.19}$ | $2.01_{\pm0.38}$ |
| SAC + VAE | $1.13_{\pm0.14}$ | $5.30_{\pm0.68}$ | $10.00_{\pm0.00}$ |
| SAC + SB | $0.98_{\pm0.88}$ | $2.08_{\pm0.30}$ | $10.00_{\pm0.00}$ |
| DreamerV3 | $\mathbf{0.42_{\pm0.06}}$ | $\mathbf{1.23_{\pm0.20}}$ | $\mathbf{1.44_{\pm0.58}}$ |
| DreamerV3 w/o dec. | $1.13_{\pm0.12}$ | $1.79_{\pm0.61}$ | $2.57_{\pm0.91}$ |

encoders in this setup, we evaluate PPO with two types of pretrained encoders. It is important to acknowledge that direct comparisons of sample efficiency with encoders trained from scratch can be nuanced, as pretrained encoders have inherently been exposed to significantly more data during their pretraining phase. The in-distribution pretrained encoder (PT (ID)) for PPO, intended as a proxy to estimate an approximate upper bound on performance achievable with pretraining, significantly boosts sample efficiency across all tested pool sizes. The out-of-distribution pretrained encoder (PT (OOD)), designed to represent the average performance one might expect from a generally pretrained encoder, offers more modest gains which also decrease with larger pools. Notably, however, the rate at which performance degrades as the image pool size increases appears to be similar regardless of whether pretraining is used or not.

For SAC, data augmentation with RAD consistently improves efficiency, especially as visual diversity increases. In contrast, auxiliary methods like CURL, DBC, VAE, SPR, and SB require more samples than standard SAC, particularly with larger pools. The underperformance of these methods appears to stem from different factors. CURL may struggle because enforcing similarity between augmented observations could be ineffective when images from the pool lack shared structure. DBC might perform poorly when observations with similar visual features correspond to different dynamics and rewards. SPR, SB, and VAE may also be less effective because enforcing latent space smoothness could be unsuited to SPGym's highly discontinuous observations, as suggested by the relatively better performance of the unconstrained AE variant.

Finally, DreamerV3 demonstrates particularly strong performance, consistently outperforming both PPO and SAC variants across all pool sizes. The full DreamerV3 model

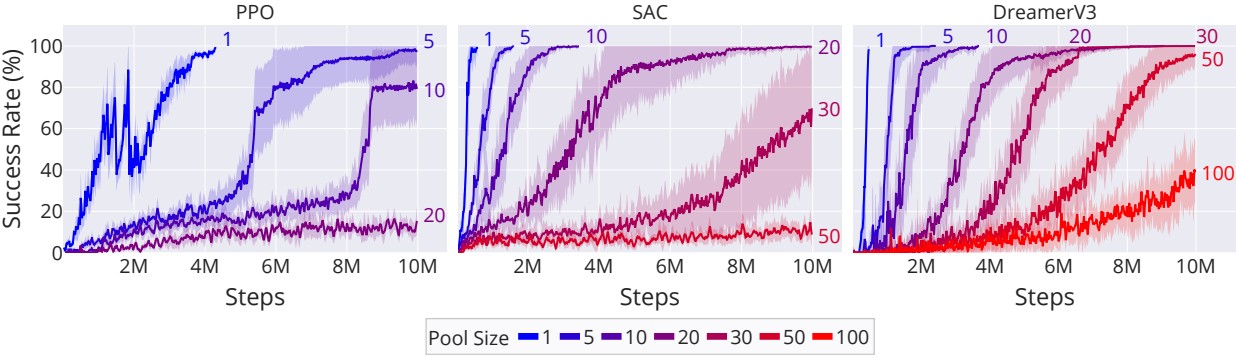

*Figure 4.* **Success rate as a function of environment steps.** The gradual increase in visual diversity affects the sample efficiency of standard PPO, SAC, and DreamerV3 agents at different rates. Each line represents a different pool size, from 1 to 100 images.

shows remarkably stable performance across different pool sizes, significantly better than most other methods. The variant without decoder gradients shows reduced performance across all configurations, highlighting the importance of the world model's discrete reconstruction objective. This suggests that DreamerV3's approach of learning a predictive model of the environment provides a particularly effective foundation for handling visual diversity in SPGym.

These results highlight SPGym's value as a diagnostic tool, effectively distinguishing which approaches can handle its distinct combination of visual diversity and structured dynamics. The benchmark exposes important limitations in methods that have proven successful in other visual RL domains, suggesting that their underlying assumptions may not align with SPGym's unique characteristics. For detailed performance analysis and proposed explanations for agent behavior, see Appendix C.2.1.

### 5.2. How does visual diversity affect agents?

Figure 4 exposes critical limitations in current methods. While all agents eventually degrade with larger image pools, their failure modes differ. PPO degrades significantly at pool size 10 and fails at 20. SAC performs well up to pool size 20, degrades at 30, and fails at 50. DreamerV3 shows the most robust scaling, learning effectively at pool size 50 and exhibiting learning even at pool size 100.

These distinct degradation patterns suggest differences in handling increasing visual diversity, potentially due to agents memorizing features instead of learning generalizable representations, thus exhausting network capacity. PPO's early failure indicates difficulty forming robust representations and rapid capacity exhaustion. SAC's better performance on medium pools, possibly aided by its replay buffer and encoder training independent of actor gradients, suggests stronger representation learning initially. How-

ever, it also faces fundamental limitations, potentially as memorization becomes unsustainable. DreamerV3's world model architecture appears to foster more compressed representations, leading to more graceful degradation, but it too struggles with very large pools.

Preliminary DreamerV3 experiments with extremely large image pools (10,000-20,000 images) show that agents fail to learn useful policies, as each observation becomes virtually unique. This suggests that simply increasing dataset scale is not enough if the RL signal becomes too sparse to effectively guide encoder training in the presence of high diversity. These methods appear to memorize rather than generalize, struggling when network capacity is overwhelmed by constantly novel inputs. This highlights fundamental limitations of current approaches when faced with diverse visuals where assumptions break and the learning signal is too weak to support meaningful learning.

### 5.3. Can agents generalize to unseen visual inputs?

Our analysis of generalization capabilities on PPO and SAC agents reveals a stark contrast between in-distribution and out-of-distribution test performance. We evaluate two levels of generalization difficulty: 'Easy' OOD, using augmented versions of training images, and 'Hard' OOD, using completely unseen pictures. Despite most agents achieving high success rates on their training distribution, they exhibit varying degrees of failure on both generalization challenges. Since meaningful generalization evaluation requires agents that first achieve competent in-distribution performance, we primarily focus our OOD analysis on smaller pool sizes (i.e., 1, 5, and 10) where most methods successfully learn the task. Full OOD performance data for both Easy and Hard settings across all configurations can be found in Appendix C.2.

For Easy OOD evaluation, we measure the success rate of PPO and SAC agents after applying each augmenta-

*Table 2.* **Success rate of PPO and SAC agents on Easy OOD across different training image pool sizes.** Higher is better.

| Algorithm | Pool 1 | Pool 5 | Pool 10 |
|---|---|---|---|
| PPO | $0.49_{\pm 0.13}$ | $0.53_{\pm 0.14}$ | $0.34_{\pm 0.08}$ |
| PPO + PT (ID) | $0.33_{\pm 0.09}$ | $0.53_{\pm 0.16}$ | $0.27_{\pm 0.07}$ |
| PPO + PT (OOD) | $0.49_{\pm 0.12}$ | $0.52_{\pm 0.14}$ | $0.34_{\pm 0.08}$ |
| SAC | $0.45_{\pm 0.12}$ | $0.58_{\pm 0.12}$ | $0.46_{\pm 0.12}$ |
| SAC + RAD | $0.62_{\pm 0.15}$ | $0.42_{\pm 0.13}$ | $0.30_{\pm 0.11}$ |
| SAC + CURL | $0.76_{\pm 0.09}$ | $0.44_{\pm 0.10}$ | $0.37_{\pm 0.11}$ |
| SAC + SPR | $0.65_{\pm 0.13}$ | $0.21_{\pm 0.09}$ | $0.07_{\pm 0.04}$ |
| SAC + DBC | $0.44_{\pm 0.13}$ | $0.34_{\pm 0.13}$ | $0.13_{\pm 0.04}$ |
| SAC + AE | $0.78_{\pm 0.11}$ | $0.64_{\pm 0.16}$ | $0.55_{\pm 0.12}$ |
| SAC + VAE | $0.64_{\pm 0.15}$ | $0.30_{\pm 0.08}$ | $0.12_{\pm 0.03}$ |
| SAC + SB | $0.89_{\pm 0.08}$ | $0.65_{\pm 0.12}$ | $0.06_{\pm 0.02}$ |

tion described in Appendix A.3, averaging results over 100 episodes per augmentation. Table 2 reports these results for training pool sizes 1, 5, and 10. As the diversity of the training pool increases, we observe a general decline in OOD performance. Some SAC variants show greater robustness to augmentations than standard SAC, especially when trained with a pool size of 1, but this advantage diminishes as the pool size grows. Overall, Easy OOD performance tends to decrease for all methods as training pool size increases. Notably, there is a strong correlation (Pearson $r = -0.81$, $p = 2.5 \times 10^{-12}$) between Easy OOD success and sample efficiency across methods and pool sizes, suggesting that agents which scale well with increasing visual diversity also tend to generalize better to augmented inputs.

For Hard OOD evaluation, we test all 5 seeds of PPO, SAC, and Dreamer agents on 100 episodes using images independently sampled from the ImageNet-1k validation split. Unsurprisingly, agents almost universally fail, achieving near 0% success rates across all methods and training configurations. This stark failure on completely unseen images is a crucial diagnostic finding, exposing fundamental limitations of current end-to-end RL methods when faced with visual inputs that differ substantially from their training distribution. This finding reinforces our interpretation that agents may be memorizing specific visual patterns instead of learning generalizable representations.

Intuitively, one might expect training on larger, more diverse image pools to improve generalization. However, our results demonstrate that even agents trained on pools of up to 100 images completely fail to transfer skills to novel visuals in the Hard OOD setting. Furthermore, performance on augmented training images often decreases as training pool size increases. This perhaps counter-intuitive result may suggest that, in SPGym, agents trained on smaller, less diverse pools, learn representations more attuned to the specific structural invariances of the task, making them robust to simple perturbations but still not fundamentally general.

The rare non-zero success rates in Hard OOD likely stem from chance encounters with nearly-solved initial states rather than genuine generalization.

These findings reveal fundamental limitations in current visual RL methods and point toward specific research directions. The universal failure on Hard OOD, combined with degradation on Easy OOD as pool diversity increases, indicates that memorization-based learning is insufficient for visual generalization. Future research should focus on developing techniques that not only achieve good sample efficiency but also explicitly promote generalization to truly novel visual contexts. This might involve architectures that better separate visual representation learning from policy learning, incorporate stronger inductive biases for visual reasoning, or leverage self-supervised objectives that encourage learning of more fundamental visual features, along with regularization methods that explicitly discourage memorization. Simply increasing the diversity of training images, as SPGym allows, is insufficient with current algorithms, highlighting the need for new approaches.

### 5.4. Is representation quality linked with performance?

As a proxy for the quality of learned representations, we performed linear probing on frozen encoders from trained PPO and SAC agents, using a single-layer MLP to predict one-hot puzzle states. We find a statistically significant correlation between probe accuracy and sample efficiency (Pearson $r = -0.81$, $p = 1.1 \times 10^{-13}$), indicating that encoders capturing more task-relevant spatial information are strongly linked with faster learning. As image pool size increases, both probe accuracy and task performance systematically degrade, with standard SAC maintaining high accuracy (100% at pool size 1, 97.63% at pool size 5) mirroring its strong efficiency, while less efficient methods like SAC+VAE (78.21% at pool size 5) and SPR (dropping from 94.31% to 75.48% from pool size 5 to 10) show reduced probe performance. These consistent trends across algorithms demonstrate SPGym's ability to identify learning procedures that develop representations better aligned with the task's spatial reasoning needs. Detailed probing results are provided in Appendix C.1.2.

### 5.5. Does performance generalize across image sources?

To validate that our findings generalize beyond ImageNet, we evaluated agents on DiffusionDB (Wang et al., 2023), a dataset of procedurally generated images. As shown in Figure 8 in Appendix A.5, performance scaling patterns on DiffusionDB closely mirror those observed on ImageNet across PPO, SAC, and DreamerV3. This consistency across fundamentally different image sources (real photographs versus synthetic generations) demonstrates that visual diversity rather than semantic content drives the representation

learning challenge in SPGym. The similarity in degradation patterns as pool size increases indicates that our algorithmic insights reflect fundamental properties of the tested methods rather than dataset-specific artifacts. Procedurally generated images also offer practical advantages for future research: eliminating storage requirements through on-demand generation, enabling fine-grained control over visual similarity, and providing unlimited training diversity.

### 5.6. How does puzzle size affect learning performance?

Beyond visual diversity, we also explore the impact of increasing puzzle size. As shown in Table 3, the complexity increase from $3 \times 3$ to $4 \times 4$ grids significantly impacts learning. On the simpler $3 \times 3$ puzzle, PPO solved the puzzle in 1.75M steps, while SAC and DreamerV3 were more efficient, requiring 0.33M and 0.42M steps, respectively. For the $4 \times 4$ puzzle, PPO's sample requirements surged to 24.46M steps, far exceeding the 10M step training budget, while SAC and DreamerV3 still solved the puzzle within budget at 8.14M and 2.26M steps, respectively. This demonstrates that while larger state spaces pose a major challenge by requiring more exploration and more complex visual representations, more sample-efficient algorithms can still scale to harder tasks.

*Table 3.* **Million steps to reach 80% success rate across grid sizes, with pool size 1.** Lower is better.

| Grid Size | PPO | SAC | DreamerV3 |
|---|---|---|---|
| 3×3 | $1.75_{\pm 0.44}$ | $0.33_{\pm 0.07}$ | $0.42_{\pm 0.06}$ |
| 4×4 | $24.46_{\pm 7.58}$ | $8.14_{\pm 3.64}$ | $2.26_{\pm 0.29}$ |

### 5.7. How optimal are the learned solutions?

While our primary focus is on sample efficiency for task completion, we also analyze solution quality by examining the average number of steps agents take to solve puzzles. Our experimental design uses early termination when agents achieve 100% success rate to enable out-of-distribution evaluation before extreme encoder overfitting and to save computational resources. However, this approach may prevent agents from discovering more optimal solutions through continued training. To investigate this trade-off, we trained PPO, SAC, and DreamerV3 on pool size 1 for the full 10M steps without early termination across 5 seeds. Comparing episode lengths between when these agents first achieve 100% success rate (where early termination would occur) and after completing the full training reveals substantial improvements in solution efficiency. For PPO, the first 100 successful episodes average 214.30 ± 16.52 steps, while the last 100 episodes average 31.35 ± 6.59 steps. SAC shows improvement from 64.16 ± 9.81 to 57.27 ± 12.29 steps, and DreamerV3 improves from 126.02 ± 17.25 to 23.48 ± 0.71 steps. Notably, DreamerV3 with continued training approaches the theoretical 22-step average optimal solution (Reinefeld, 1993). This confirms that early termination, while needed for our experimental objectives, prevents agents from discovering more optimal solutions through continued training.

## 6. Conclusion

We introduce the Sliding Puzzles Gym (SPGym), a novel benchmark designed to systematically evaluate representation learning in RL algorithms by isolating visual complexity controls from the environment dynamics. Our analysis reveals fundamental tensions challenging current visual RL methods. We show that sophisticated representation learning techniques struggle with SPGym's combination of visual diversity and structured spatial reasoning requirements, with auxiliary objectives often underperforming simpler approaches like data augmentation. We also uncover a complex relationship between sample efficiency and generalization: agents trained on smaller pools learn task-specific invariances that paradoxically make them more robust to simple perturbations, with larger pools often degrading generalization performance. Most critically, we expose severe limitations in end-to-end RL methods through universal failure on novel visual contexts, achieving near-zero success despite strong in-distribution performance – revealing that learned representations rely on memorization rather than genuine visual understanding. These findings highlight the need for algorithmic advances that move beyond memorization toward true visual understanding.

**Limitations.** We identify two key limitations in this work. First, our objective was to evaluate out-of-the-box performance of all methods with minimal tuning, which means we may not have seen the true peak capabilities of each approach. Second, computational constraints limited us to 5 independent runs per configuration. Given the high stochasticity in sampled image pools, more seeds would provide better statistical robustness for comparing methods.

**Future work.** SPGym opens several promising research directions for developing representation learning methods that can bridge the generalization gap while maintaining sample efficiency. The benchmark's controlled nature enables systematic investigation of the trade-offs between training diversity and generalization capability, while its easy integration with external image generation systems allows controlled experiments with varying degrees of visual similarity. Beyond images, SPGym supports incorporating other modalities as puzzle observations, enabling research on leveraging powerful pretrained models as representation encoders. We believe these directions would help develop more robust learning approaches that transfer effectively across different domains and modalities.

## Acknowledgements

The authors gratefully acknowledge the valuable insights and constructive discussions provided by Professors Marcos R. O. A. Maximo and Flávio H. T. Vieira. We also thank the reviewers for their thoughtful feedback and constructive suggestions that helped improve this paper.

This work has been partially funded by the project Research and Development of Digital Agents Capable of Planning, Acting, Cooperating and Learning supported by Advanced Knowledge Center in Immersive Technologies (AKCIT), with financial resources from the PPI IoT/Manufatura 4.0 / PPI HardwareBR of the MCTI grant number 057/2023, signed with EMBRAPII.

Luckeciano C. Melo acknowledges funding from the Air Force Office of Scientific Research (AFOSR) European Office of Aerospace Research & Development (EOARD) under grant number FA8655-21-1-7017.

## Impact Statement

This work advances our understanding of how visual representation learning affects reinforcement learning performance and generalization. The insights gained could influence the development of more robust and efficient RL systems for real-world applications. However, several potential impacts warrant consideration:

**Positive impacts:** By exposing limitations in current methods, our benchmark may help direct research toward more reliable and generalizable RL systems. This could accelerate progress in areas like robotics and autonomous systems where visual understanding is crucial. The benchmark's controlled nature also promotes more rigorous evaluation practices in RL research.

**Negative impacts:** As with any machine learning benchmark, there is a risk of overfitting to our specific evaluation criteria rather than addressing fundamental challenges. Additionally, improved visual RL capabilities could enable automation that displaces human workers or enable surveillance applications if misused.

**Mitigations:** We have open-sourced our benchmark and evaluation protocols to promote transparency and reproducibility. We encourage future work to consider both performance metrics and broader societal implications when building on our findings. Researchers should carefully consider potential dual-use applications and implement appropriate safeguards when deploying systems based on these techniques.

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

# A. Experimental Setup and Configuration

## A.1. Model Architectures

We base our implementations on CleanRL's Atari agents for both PPO and SAC, with minor architectural modifications including additional normalization layers and increased network depth to approximate our SAC implementation to the one used by Tomar et al. (2023). The architectures are detailed below.

For PPO agents:

```
SharedEncoder(
    (encoder): Sequential(
        (0): Conv2d(3, 32, kernel_size=(8, 8), stride=(4, 4))
        (1): ReLU()
        (2): BatchNorm2d(32, eps=1e-05, momentum=0.1, affine=True, stats=True)
        (3): Conv2d(32, 64, kernel_size=(4, 4), stride=(2, 2))
        (4): ReLU()
        (5): BatchNorm2d(64, eps=1e-05, momentum=0.1, affine=True, stats=True)
        (6): Conv2d(64, 64, kernel_size=(3, 3), stride=(1, 1))
        (7): BatchNorm2d(64, eps=1e-05, momentum=0.1, affine=True, stats=True)
        (8): Flatten(start_dim=1, end_dim=-1)
    )
    (projection): Sequential(
        (0): Linear(in_features=3136, out_features=512, bias=True)
        (1): LayerNorm((512,), eps=1e-05, elementwise_affine=True)
        (2): Tanh()
    )
)
Actor(
    (encoder): SharedEncoder
    (mlp): Linear(in_features=512, out_features=4, bias=True)
)
Critic(
    (encoder): SharedEncoder
    (mlp): Linear(in_features=512, out_features=1, bias=True)
)
```

For SAC agents:

```
SharedEncoder(
    (encoder): Sequential(
        (0): Conv2d(3, 32, kernel_size=(8, 8), stride=(4, 4))
        (1): ReLU()
        (2): BatchNorm2d(32, eps=1e-05, momentum=0.1, affine=True, stats=True)
        (3): Conv2d(32, 64, kernel_size=(4, 4), stride=(2, 2))
        (4): ReLU()
        (5): BatchNorm2d(64, eps=1e-05, momentum=0.1, affine=True, stats=True)
        (6): Conv2d(64, 64, kernel_size=(3, 3), stride=(1, 1))
        (7): BatchNorm2d(64, eps=1e-05, momentum=0.1, affine=True, stats=True)
        (8): Flatten(start_dim=1, end_dim=-1)
    )
)
Actor(
    (encoder): Encoder(
        (shared_encoder): SharedEncoder
        (projection): Sequential(
            (0): Linear(in_features=3136, out_features=512, bias=True)
            (1): LayerNorm((512,), eps=1e-05, elementwise_affine=True)
            (2): Tanh()
        )
    )
    (mlp): Sequential(
        (0): Linear(in_features=512, out_features=512, bias=True)
        (1): LayerNorm((512,), eps=1e-05, elementwise_affine=True)
        (2): ReLU()
```

```
            (3): Linear(in_features=512, out_features=4, bias=True)
        )
)
Critic(
    (encoder): Encoder(
        (shared_encoder): SharedEncoder
        (projection): Sequential(
            (0): Linear(in_features=3136, out_features=512, bias=True)
            (1): LayerNorm((512,), eps=1e-05, elementwise_affine=True)
            (2): Tanh()
        )
    )
    (mlp): Sequential(
        (0): Linear(in_features=512, out_features=512, bias=True)
        (1): LayerNorm((512,), eps=1e-05, elementwise_affine=True)
        (2): ReLU()
        (3): Linear(in_features=512, out_features=4, bias=True)
    )
    (mlp): Sequential(
        (0): Linear(in_features=512, out_features=512, bias=True)
        (1): LayerNorm((512,), eps=1e-05, elementwise_affine=True)
        (2): ReLU()
        (3): Linear(in_features=512, out_features=4, bias=True)
    )
)
```

For the representation learning methods, we maintain the same base architecture and use a consistent MLP structure (2 layers with ReLU activation) for the projector, predictor, transition and reward models. The decoder architecture is as follows:

```
ImageDecoder(
    (decoder): Sequential(
        (0): Linear(in_features=512, out_features=512, bias=True)
        (1): LayerNorm((512,), eps=1e-05, elementwise_affine=True)
        (2): ReLU()
        (3): Linear(in_features=512, out_features=3136, bias=True)
        (4): Unflatten(dim=1, unflattened_size=(3, 7, 7))
        (5): ConvTranspose2d(3, 64, kernel_size=(3, 3), stride=(1, 1))
        (6): ReLU()
        (7): BatchNorm2d(64, eps=1e-05, momentum=0.1, affine=True, stats=True)
        (8): ConvTranspose2d(64, 32, kernel_size=(4, 4), stride=(2, 2))
        (9): ReLU()
        (10): BatchNorm2d(32, eps=1e-05, momentum=0.1, affine=True, stats=True)
        (11): ConvTranspose2d(32, 3, kernel_size=(8, 8), stride=(4, 4))
        (12): Sigmoid()
    )
)
```

For DreamerV3 agents, we use the same base 12M architecture as the one used by Hafner et al. (2025). Appendix A.2 contains the specific hyperparameters used in our experiments.

### A.2. Hyperparameters

Table 5, Table 6, and Table 7 list hyperparameters used across all experiments, unless noted otherwise. For DreamerV3, we adopted hyperparameters from (Hafner et al., 2025), modifying only the decoder loss scale (set to 0) for the version without decoder. Table 8 lists hyperparameters for representation learning methods and components. We use a separate optimizer for the representation learning gradient flow, and we adopt a higher learning rate. When using representation learning methods, we update the target network's encoder faster, with an EMA $\tau$ of 0.025. When using a crop augmentation, we set the image size to 100 and crop it back to 84.

*Table 4.* **Benchmark settings**

| Parameter | Value |
| --- | --- |
| Max steps | 10M |
| Puzzle size | 3x3 |
| Action space | discrete |
| Variation | image |
| Render size | 100x100 (for crop augmentation) |
| | 84x84 (otherwise) |
| Dataset | ImageNet-1k validation split |

*Table 5.* **Hyperparameters for PPO**

| PPO Parameter | Value |
| --- | --- |
| Input image size | 84x84 |
| Env instances | 64 |
| Optimizer | Adam |
| Learning Rate (LR) | 2.5e-4 |
| LR annealing | yes |
| Adam $\epsilon$ | 1e-5 |
| Num. steps | 16 |
| Num. epochs | 4 |
| Batch size | 64 |
| Num. minibatches | 4 |
| $\gamma$ | 0.99 |
| GAE $\lambda$ | 0.95 |
| Advantage normalization | yes |
| Clip coef. | 0.1 |
| Clip value loss | yes |
| Value function coef. | 0.5 |

*Table 6.* **Hyperparameters for SAC**

| SAC Parameter | Value |
| --- | --- |
| Input image size | 84x84 |
| Env instances | 64 |
| Optimizer | Adam |
| Learning rate | 3e-4 |
| Replay buffer capacity | 3e5 |
| Batch size | 4096 |
| Warmup steps | 2e4 |
| $\gamma$ | 0.99 |
| Policy update frequency | 2 |
| Fixed $\alpha$ temperature | 0.05 |
| Target network update frequency | 1 |
| Target Q functions EMA $\tau$ | 0.005 |
| Target encoder EMA $\tau$ | 0.005 (standard) |
| | 0.025 (otherwise) |

*Table 7.* **Hyperparameters for DreamerV3**

| DreamerV3 Parameter | Value |
|---|---|
| Input image size | 80x80 |
| Env instances | 16 |
| Model size | 12M |
| RSSM deterministic size | 2048 |
| RSSM hidden size | 256 |
| RSSM classes | 16 |
| Network depth | 16 |
| Network units | 256 |
| Replay buffer capacity | 5e5 |
| Replay ratio | 32 |
| Action repeat | 1 |
| Learning rate | 4e-5 |
| Batch size | 16 |
| Batch length | 64 |
| Imagination horizon | 15 |
| Discount horizon | 333 |
| Decoder loss scale | 1 (standard) |
| | 0 (no decoder) |

*Table 8.* **Hyperparameters for Representation Learning Methods and Components**

| Method | Hyperparameter | Value |
|---|---|---|
| General | Learning rate | 1e-3 |
| | Loss coefficient | 1.0 |
| Transition and reward models | Min sigma | 1e-4 |
| | Max sigma | 10 |
| | Probabilistic | no |
| SAC-AE | Latent space decay weight | 1e-6 |
| | Decoder decay weight | 1e-7 |
| SAC-VAE | Variational KL weight $\beta$ | 1e-7 |
| CURL | Temperature | 0.1 |
| | Positive samples | temporal/augmented |
| | Augmentations | crop |
| RAD | Augmentations | crop |
| | | channel_shuffle |
| | | color_jitter |
| SPR | Horizon $H$ | 5 |
| | Augmentations | crop |

### A.3. Augmentation Strategies

We evaluated several image augmentations in our preliminary experiments, as described in Appendix A.4.2. These augmentations are illustrated in Figure 5, and are as follows:

- **No augmentation**: The image is fed as is to the agent.

- **Crop**: Randomly crops a portion of the image and resizes it back to the original dimensions. This helps learn translation invariance by forcing the agent to recognize patterns regardless of their position in the frame.

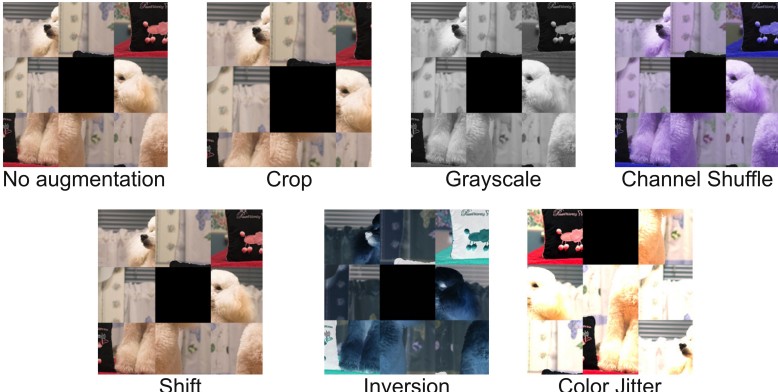

*Figure 5.* **Example of image augmentations.** The top left image shows the original observation, and the subsequent images show the observation after each augmentation procedure is independently applied.

- **Grayscale**: Converts the RGB image to grayscale by averaging across color channels. This reduces the visual complexity and helps the agent focus on structural features rather than color information.

- **Channel Shuffle**: Randomly permutes the RGB color channels. This encourages the agent to be invariant to color transformations while preserving the image structure.

- **Shift**: Translates the image by a small random amount in both horizontal and vertical directions. Similar to crop, this promotes translation invariance in the learned representations.

- **Inversion**: Inverts the pixel values by subtracting them from the maximum possible value (255 for 8-bit images). This teaches the agent to recognize patterns independent of absolute intensity values.

- **Color Jitter**: Applies random color variations to the image, including brightness, contrast, saturation, and hue. This helps the agent to be invariant to color transformations while preserving the image structure.

After extensive experimentation, we found that the combination of grayscale and channel shuffle consistently produced the best results. This combination effectively reduces visual complexity while maintaining important structural information. We adopted this augmentation pair as the standard for all our agents that use augmentation techniques.

### A.4. Preliminary Experiments and Design Rationale

#### A.4.1. HYPERPARAMETER SELECTION PROCESS

For SAC agents, the entropy coefficient $\alpha$ significantly impacts performance (Haarnoja et al., 2018a). While Haarnoja et al. (2018b) proposed automatic tuning (autotune) based on policy entropy, we found this approach ineffective for SPGym, even with various learning rates (LRs). Through systematic Hyperband (Li et al., 2018) sweeps over pools of size 1, we identified $\alpha = 0.05$ as optimal (Figure 6). This value provides a good balance between exploration and exploitation, allowing the agent to efficiently learn the puzzle mechanics while maintaining enough randomness to discover new solutions.

PPO and DreamerV3 agents proved more robust to hyperparameter choices, performing well with their default configurations. This robustness is particularly valuable in our benchmark setting, as it suggests these algorithms

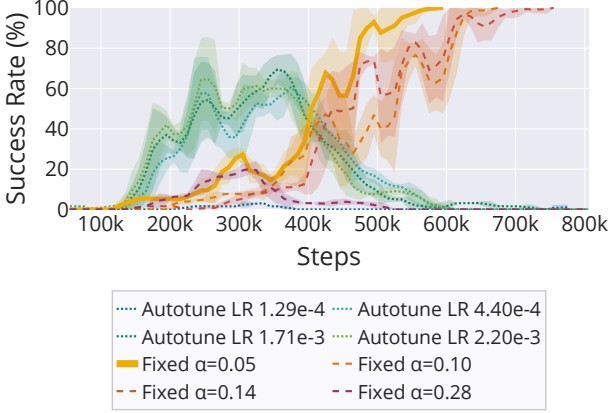

*Figure 6.* **Performance comparison of different $\alpha$ values for SAC agents on pool size 1.** Fixed $\alpha = 0.05$ outperforms automatic tuning approaches across different learning rates.

can adapt to new tasks without extensive tuning. Complete hyperparameter settings are provided in Appendix A.2.

### A.4.2. DATA AUGMENTATION ANALYSIS AND CHOICE

Building on insights from Laskin et al. (2020a) that environment-specific invariances influence optimal augmentation strategies, we systematically evaluated augmentation pipelines for RAD, CURL, and SPR in SPGym. Our analysis focused on sample efficiency to maintain consistency with our core experimental objectives. We tested the five augmentation techniques detailed on Appendix A.3 on pools of 5 images, which offered a good balance between task complexity and convergence speed. For SPR, specifically, we also experimented with shift + color jitter, as suggested by Schwarzer et al. (2020). Across all algorithms, our experiments consistently converged to a simple two-stage augmentation process: probabilistic grayscale conversion (with 20% chance) followed by channel shuffling (Figure 7). This pipeline's effectiveness likely stems from its ability to simultaneously reduce visual complexity through grayscale conversion while introducing beneficial stochasticity via channel shuffling, with both transformations preserving critical structural information while preventing overfitting to specific color patterns. We adopted this augmentation combination for all subsequent experiments. While our current evaluation focused on sample efficiency, investigating how different augmentation strategies affect generalization remains an important direction for future work.

### A.5. Dataset Analysis and Choice

Our choice of ImageNet-1k as the primary dataset for SPGym was motivated by several key considerations. First, ImageNet provides a diverse set of real-world images that challenge agents to learn generalizable visual representations. We hypothesized that real-world images would provide unique insights related to representation learning that should be applicable to other domains beyond the puzzle proposed in SPGym, as they contain the complex visual patterns and structures that agents encounter in practical applications. However, as shown in Figure 8 and in comparison to Figure 4, the performance scaling patterns we observe on ImageNet closely mirror those on DiffusionDB, suggesting that our findings are not specific to a particular dataset but rather reflect fundamental properties of the algorithms being tested.

The similarity in scaling behavior between ImageNet and DiffusionDB is particularly noteworthy because these datasets differ substantially in their composition and generation process. While ImageNet consists of real photographs, DiffusionDB contains synthetic images generated by text-to-image models. The consistent performance patterns across these datasets suggest that our results capture fundamental algorithmic behaviors rather than dataset-specific artifacts.

Our demonstration with DiffusionDB reveals promising directions for future work with procedurally generated datasets. As shown in our analysis, agents perform similarly on ImageNet and DiffusionDB, suggesting that visual diversity rather than semantic content drives difficulty. Procedurally generated images offer several compelling advantages worth further investigation: they eliminate the need for large image storage by generating unique images on demand for each episode, enable fine-grained control over the generalization challenge by gradually increasing visual differences between generated images, and provide virtually unlimited training data diversity. These capabilities could enable more systematic studies of

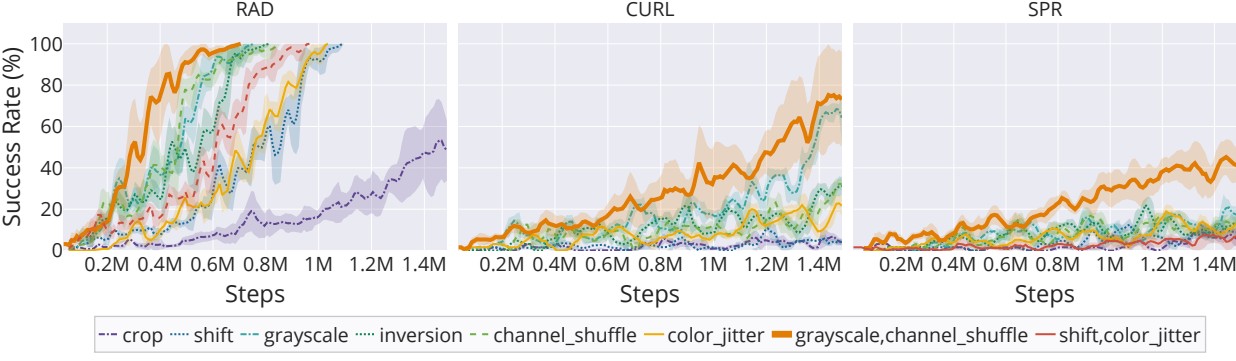

*Figure 7.* **Comparative analysis of data augmentation strategies.** Results show performance of SAC with RAD, CURL and SPR on 5-image pools. Grayscale conversion and channel shuffling emerge as the most effective combination, significantly outperforming other augmentation strategies.

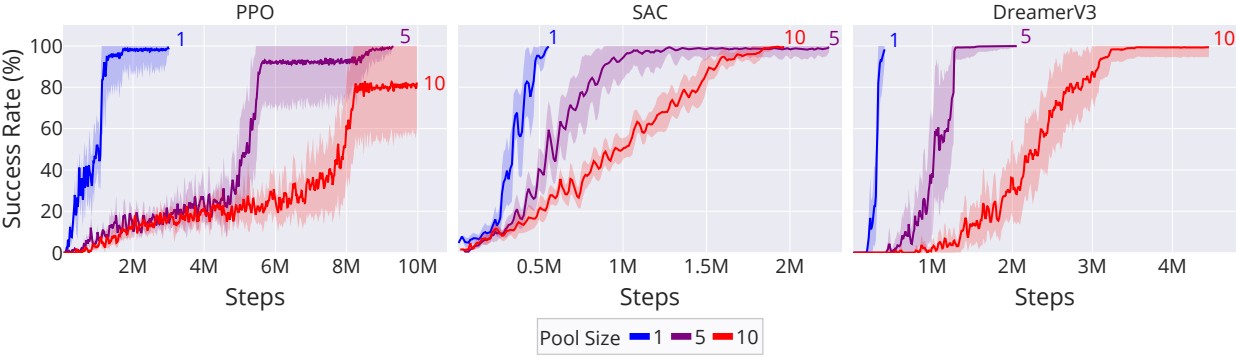

*Figure 8.* **Performance scaling with DiffusionDB images.** Success rates for PPO, SAC, and DreamerV3 agents across different pool sizes (1, 5, and 10) using DiffusionDB images. The performance patterns closely mirror those seen with ImageNet (Figure 4). Shaded regions represent 95% confidence intervals across 5 independent seeds.

visual generalization in reinforcement learning.

This cross-dataset consistency strengthens our confidence in the generalizability of our findings. It indicates that the relative performance of different representation learning methods is driven more by their core assumptions and architectural choices than by the specific characteristics of the training data. This is particularly important for our goal of understanding how different approaches handle increasing visual diversity, as it suggests our conclusions may extend to other domains beyond the specific datasets used in our experiments.

### A.6. Hardware Setup and Runtime

Our hardware setup consists of an AMD Ryzen 7 3700X CPU, an NVIDIA RTX 3090 GPU, 64GB of RAM, and 128GB of swap space. Using this configuration, DreamerV3 experiments take approximately 20 hours per run, primarily because of the heavy use of swap space for replay buffers, which must store hundreds of thousands of images in memory. SAC takes between 2 (e.g. standard, RAD) to 11 hours (e.g. SPR, SB) depending on the representation learning components used. For SAC, the longer runtimes are due to the number of sequential inference steps required to train the agent and auxiliary networks. PPO experiments are significantly faster, with the longest runs completing in about 1 hour and 30 minutes.

## B. Algorithms and Variations

We describe the algorithms and their variations used in this work in detail. Our experiments employ three main algorithms – PPO, SAC, and DreamerV3 – each with different representation learning approaches. We focus particularly on how these methods process and learn from visual observations, as this is crucial for performance in our benchmark.

### B.1. Pretraining and PPO

Drawing inspiration from previous work on pretraining methods in RL (Higgins et al., 2017; Stooke et al., 2021; Schwarzer et al., 2021), we implement a pretraining approach for PPO that focuses on learning task-relevant visual representations. The pretraining process involves training a PPO agent to completion on a single environment instance, then extracting its CNN weights. These pretrained weights are then used to initialize new PPO agents, while all other network components (policy and value networks) start from random initialization. We evaluate two scenarios: in-distribution (ID), where new agents are trained on the same pool of images used during pretraining, and out-of-distribution (OOD), where a different image pool is sampled. The ID setting represents an upper bound on what pretraining can achieve with perfect visual alignment, while the OOD setting reflects the more realistic scenario of deploying pretrained encoders on novel visual inputs, similar to how general-purpose pretrained models would be used in practice.

## B.2. Data Augmentation and RAD

Reinforcement Learning from Augmented Data (RAD) (Laskin et al., 2020a) represents a simple yet effective approach to visual representation learning in RL. The key insight is that applying data augmentation to observations can improve sample efficiency by exposing the agent to transformed versions of experienced states. This creates an implicit regularization effect that helps learn more robust representations.

In our implementation, we combine RAD with SAC and apply data augmentation consistently during both policy updates and value function learning. These augmentations are applied to observations sampled from the replay buffer before being processed by the encoder network. The augmentation pipeline consists of two key transformations identified through our preliminary experiments (Appendix A.4.2): grayscale conversion and channel shuffling. Following the SPR approach (Schwarzer et al., 2020), we apply augmentations independently to each transition after sampling batches from the replay buffer, meaning that samples from the same episode may be augmented differently. We detail how each augmentation procedure is implemented in Appendix A.3.

## B.3. Contrastive Learning and CURL

Contrastive learning methods learn representations by maximizing similarity between different views of the same observation while minimizing similarity to other observations. In the context of RL, CURL (Laskin et al., 2020b) applies this principle by using data augmentation to create positive pairs, enabling agents to learn invariant representations. The method achieves this by applying random crops to observations and treating differently augmented views of the same observation as positive pairs.

The contrastive loss for a positive pair of observations $(x, x^+)$ and a set of negative examples $\{x_i^-\}$ is formulated as:

$$\mathcal{L}_{CURL} = -\log \frac{\exp(f_\theta(x)^T f_\theta(x^+)/\alpha)}{\exp(f_\theta(x)^T f_\theta(x^+)/\alpha) + \sum_i \exp(f_\theta(x)^T f_\theta(x_i^-)/\alpha)}, \tag{3}$$

where $\alpha$ is a temperature parameter and $f_\theta$ is the encoder function.

In our implementation, we combine CURL with SAC and use the same augmentation strategy identified in Appendix A.3 (grayscale conversion and channel shuffling) rather than the random crops from the original CURL paper. The encoder is trained jointly with the RL objective, allowing the representations to adapt to both the contrastive learning task and the control problem. Negative examples are drawn from other observations within the same batch, providing a computationally efficient way to obtain contrastive pairs without requiring additional memory storage.

## B.4. State Metrics and DBC

Deep Bisimulation for Control (DBC) (Zhang et al., 2021) takes a different approach to representation learning by focusing on behavioral similarity between states rather than visual similarity. The key idea is to learn an encoder that maps states to a representation space where distances reflect how similarly states behave in terms of rewards and transitions, rather than how visually similar they appear.

Given pairs of observations $(x_i, x_j)$, DBC trains an encoder $f_\theta$ to minimize:

$$J(\phi) = \left( \|\hat{z}_i - \hat{z}_j\|_1 - |r_i - r_j| - \gamma W_2(\hat{\mathcal{P}}(\cdot|\bar{z}_i, a_i), \hat{\mathcal{P}}(\cdot|\bar{z}_j, a_j)) \right)^2, \tag{4}$$

where $\hat{z}_i = f_\theta(x_i)$ represents the encoded state, $\bar{z}_i = sg(f_\theta(x_i))$ is the stop-gradient version of the encoding, and $\hat{\mathcal{P}}$ is a probabilistic transition model that predicts the next state distribution. The $W_2$ term represents the 2-Wasserstein distance between predicted transition distributions, which for Gaussian distributions has a closed-form solution (Zhang et al., 2021).

In our implementation, we combine DBC with SAC, jointly training the encoder with both the bisimulation objective and the RL objective. The transition model operates in latent space, predicting Gaussian distributions over next states. This approach helps the agent learn representations that capture behaviorally meaningful features while ignoring visual distractors that don't affect the game dynamics. Unlike methods that rely on data augmentation or reconstruction, DBC's focus on behavioral similarity makes it particularly suited for environments where visually different states might require similar actions.

## B.5. Reconstruction-Based Methods and SAC-AE/VAE

Reconstruction-based methods learn representations by training an encoder-decoder architecture to compress and reconstruct observations. We evaluate two variants combined with SAC: SAC-AE using a deterministic autoencoder and SAC-VAE using a variational autoencoder (Yarats et al., 2021b).

For SAC-AE, given an observation $x$ from the replay buffer, we train an encoder $f_\theta$ and decoder $g_\phi$ to minimize:

$$\mathcal{L}_{RAE} = \mathbb{E}_{x \sim \mathcal{D}} \left[ \|x - g_\phi(f_\theta(x))\|^2 + \lambda_z \|f_\theta(x)\|_2^2 + \lambda_\phi \|\phi\|_2^2 \right], \tag{5}$$

where $\lambda_z$ and $\lambda_\phi$ are regularization coefficients that help prevent representation collapse and overfitting respectively (Ghosh et al., 2020).

For SAC-VAE, we replace the deterministic encoder with a probabilistic encoder $q_\psi$ that outputs a distribution over latent states. The training objective becomes:

$$\mathcal{L}_{VAE} = \mathbb{E}_{q_\psi(\hat{z}|x)}[\log g_\phi(x|\hat{z})] - \beta D_{KL}(q_\psi(\hat{z}|x)\|\mathcal{N}(0,1)), \tag{6}$$

where $\beta$ controls the trade-off between reconstruction quality and latent space regularization.

In both variants, we train the encoder jointly with the SAC objective, allowing the representations to adapt to both reconstruction and control tasks. The encoded states are used as inputs to the policy and value networks. Unlike methods that rely on data augmentation or behavioral similarity, these approaches learn representations by explicitly modeling the visual structure of observations through reconstruction.

## B.6. World Models and DreamerV3

World models learn to predict future states and outcomes by learning a compact latent representation of the environment. DreamerV3 (Hafner et al., 2025) represents the state-of-the-art in world model-based reinforcement learning, employing an online encoder $f_\theta$ that maps observations $x_t$ into latent states $\hat{z}_t$. A recurrent dynamics model $h_\omega$ operates in this latent space to predict future states conditioned on actions, while a reward predictor estimates immediate rewards.

The model is trained using multiple objectives that create a multi-task learning pressure. The encoder and a corresponding decoder are trained to reconstruct observations, ensuring the latent space captures relevant visual features. The dynamics model is trained to predict future latent states that lead to accurate reconstructions of future observations. This temporal consistency objective forces the representations to be predictive of future states while supporting reconstruction and control.

DreamerV3 introduces several innovations for stable representation learning, including KL balancing to maintain informative latent states and symmetric cross-entropy loss for better gradients. Unlike methods focused solely on visual similarity, world models must learn representations that serve multiple purposes – capturing visual features, encoding dynamics, and providing a suitable space for policy learning. We refer readers to Hafner et al. (2025) for implementation details.

## B.7. Temporal Consistency Methods

Several methods leverage temporal consistency in the environment to learn better representations. These approaches are based on the principle that a good representation should not only capture the current state but also be predictive of future states and outcomes.

### B.7.1. SELF-PREDICTIVE REPRESENTATIONS (SPR)

SPR (Schwarzer et al., 2020) represents a non-contrastive approach that learns by predicting future latent states. Given a sequence of states and actions $(x_{t:t+K}, a_{t:t+K})$ from the replay buffer, where K is the prediction horizon, SPR employs:

- An online encoder $f_\theta$ that maps observations to latent states: $\hat{z}_t = f_\theta(x_t)$

- A target encoder $f_{\theta'}$ providing stable training targets, updated via exponential moving average

- An action-conditioned transition model $h_\omega$ that predicts future latent states: $\hat{z}_{t+k+1} = h_\omega(\hat{z}_{t+k}, a_{t+k})$

- Projection networks $p_\xi$, $p_{\xi'}$ and prediction head $w_\zeta$ that transform representations for the prediction task

The model generates predictions $\hat{y}_{t+k} = w_\zeta(p_\xi(\hat{z}_{t+k}))$ and compares them to target projections $\tilde{y}_{t+k} = p_{\xi'}(\tilde{z}_{t+k})$ using a cosine similarity loss:

$$\mathcal{L}_{\text{SPR}} = -\sum_{k=1}^{K} \left( \frac{\tilde{y}_{t+k}}{\|\tilde{y}_{t+k}\|_2} \right)^T \left( \frac{\hat{y}_{t+k}}{\|\hat{y}_{t+k}\|_2} \right) \tag{7}$$

### B.7.2. SIMPLE BASELINE METHOD

Tomar et al. (2023) take a minimalist approach to temporal consistency by combining two predictive objectives: reward prediction and transition prediction. We refer to this approach as the Simple Baseline (SB) method. While originally intended as a baseline, it demonstrates the effectiveness of basic temporal prediction for representation learning.

The method augments standard RL algorithms with two predictive components in latent space:

- A transition model $h_\omega$ that predicts the next encoded state

- A reward predictor $h_{\text{reward}}$ that estimates immediate rewards

The transition loss $\mathcal{L}_{\text{dyn}}$ measures the mean squared error between predicted and actual next encoded states, while the reward loss $\mathcal{L}_{\text{reward}}$ measures the error in reward predictions. These predictive losses are combined with the standard RL objective:

$$\mathcal{L}_{\text{total}} = \mathcal{L}_{\text{RL}} + \mathcal{L}_{\text{reward}} + \mathcal{L}_{\text{dyn}} \tag{8}$$

In our implementations, we combine both SPR and SB with SAC. For SPR, we apply the augmentation strategy identified in Appendix A.3 before encoding observations. Both methods operate entirely in latent space, avoiding the computational cost of pixel-space reconstruction while leveraging temporal structure to learn meaningful representations.

## C. Supplementary Analyses and Results

### C.1. Representation Learning Analysis

We further evaluate SPGym's representation learning assessment capabilities through two analyses: (1) comparing raw pixel vs. ground-truth state learning, and (2) linear probes of learned representations.

### C.1.1. STATE-BASED VS. IMAGE-BASED OBSERVATIONS

To establish a baseline and understand the impact of the visual representation challenge, we trained PPO, SAC, and DreamerV3 agents using SPGym's one-hot encoding variation. These one-hot vectors represent the ground-truth puzzle state, identical to the targets used for our linear probes (see Appendix C.1.2). We compared their sample efficiency (steps to 80% success, averaged over 5 seeds) against their image-based counterparts. For PPO and SAC agents processing one-hot vectors, CNN encoders were replaced with 2-layer MLPs. DreamerV3 utilized its default non-image encoder, a 3-layer MLP. Hyperparameters were kept consistent with image-based experiments, without specific tuning for the one-hot setting.

*Table 9.* **Steps to 80% success on one-hot vs. image-based observations.** Lower is better. '-' indicates experiments were not run for that specific configuration due to computational constraints.

| Algorithm | Grid Size | One-hot | Image (Pool 1) | Image (Pool 5) |
|---|---|---|---|---|
| PPO | 3x3 | 661.69k±81.44k | 1.75M±444.81k | 7.80M±1.08M |
| | 4x4 | 12.29M±467.84k | 24.46M±7.58M | - |
| SAC | 3x3 | 672.51k±63.10k | 334.26k±67.47k | 907.21k±116.20k |
| | 4x4 | 5.09M±463.14k | 8.14M±3.64M | - |
| DreamerV3 | 3x3 | 834.86k±61.10k | 417.09k±55.03k | 1.23M±199.49k |
| | 4x4 | 3.68M±436.97k | 2.26M±287.23k | 5.81M ± 2.17M |

The results in Table 9 offer several insights. For PPO (both grid sizes) and SAC (4x4 grid), learning directly from ground-truth one-hot states is more sample efficient than learning from images. This is expected, as the one-hot encoding removes

the burden of representation learning from pixels. The instances where image-based agents (SAC and DreamerV3 on 3x3 grid, pool 1) converged faster than their one-hot counterparts might be attributed to differences in network architectures (MLP vs. CNN/Transformer backbones) and the absence of specific tuning for the one-hot setting.

Crucially, the one-hot encoding setting presents a fixed, minimal representation learning challenge. In contrast, SPGym's image-based variations allow for a systematic scaling of the visual diversity challenge by increasing the image pool size (e.g., Pool 1 vs. Pool 5 vs. Pool 10, etc., see Appendix C.2). Across all agents and grid sizes, increasing visual diversity from pool size 1 to pool size 5 (and beyond) consistently increases sample complexity. This demonstrates SPGym's ability to isolate and stress the visual representation learning component, as the underlying task dynamics remain constant. This controlled evaluation reveals limitations in how effectively different RL agents learn representations under scalable visual diversity, insights not apparent from the one-hot setting alone. While perfect disentanglement of representation learning from policy learning is challenging in end-to-end training, SPGym provides a valuable framework for structured, comparative evaluation of visual representation learning capabilities in RL.

### C.1.2. LINEAR PROBE ANALYSIS

To directly assess the quality of learned visual representations, we conducted linear probing experiments. For each trained PPO and SAC agent, we froze its encoder and trained a single-layer MLP classifier on top of the features extracted by it until convergence. The classifier's task was to predict the one-hot encoded ground-truth puzzle state corresponding to the input image. This setup allows us to quantify how much task-relevant spatial information is captured by the agent's encoder. High probe accuracy indicates the encoder has learned features that are linearly separable with respect to the underlying game state.

Our analysis reveals several key insights. First, we find a strong, statistically significant correlation between linear probe accuracy and agent sample efficiency (Pearson r=-0.81, p=1.1e-13), with higher probe accuracy being highly predictive of fewer environment steps required to reach 80% success. This suggests that agents whose encoders capture more task-relevant spatial information tend to learn the task more efficiently.

Second, examining probe accuracies across pool sizes and algorithms further clarifies this relationship. As image pool size increases, both probe accuracy and task performance systematically degrade, isolating the effect of visual diversity on representation learning. For example, standard SAC maintains high probe accuracy (100% at pool size 1, 97.63% at pool size 5), mirroring its strong sample efficiency. In contrast, methods with lower sample efficiency show reduced probe performance: SAC+VAE achieves only 78.21% probe accuracy at pool size 5, while SAC+RAD reaches 98.66%. Other representation learning methods that underperform standard SAC, such as SPR and DBC, also exhibit declining probe accuracy as pool size increases (e.g., SPR drops from 94.31% at pool size 5 to 75.48% at pool size 10).

These trends indicate that different algorithms develop representations with varying alignment to the spatial reasoning demands of the task. The consistent link between probe accuracy and sample efficiency across diverse methods suggests that SPGym can help identify which learning procedures lead to representations that better support task performance.

Full linear probe accuracy data for all agents and pool sizes are presented in Table 10 below. Comprehensive performance metrics, including the sample efficiency data used in the correlation analysis, can be found in Appendix C.2.

### C.2. Detailed Performance Analysis

This section provides comprehensive analysis of algorithmic performance in SPGym, examining how different representation learning approaches handle increasing visual diversity. We present detailed learning curves showing the training dynamics of each method, along with quantitative performance metrics across varying pool sizes.

The analysis includes detailed algorithmic variant analysis explaining why certain methods succeed or fail, and comprehensive performance curves for PPO (Figure 10), SAC (Figure 11), and DreamerV3 (Figure 12). Tables 11 to 13 provide quantitative metrics per pool size (Pool) including sample efficiency (Steps), and episode length (Length), across algorithms and variants. Tables 11 and 12 also include in-distribution and out-of-distribution success rates (ID Success, OOD Easy, OOD Hard), which were not computed for DreamerV3 due to complexities in the original codebase. The rare non-zero success rates in Hard OOD likely stem from chance encounters with nearly-solved initial states rather than genuine generalization.

*Table 10.* **Linear probe accuracy (%) for each agent and pool size.** Accuracy of the linear probe indicates linearly separable features capturing task-relevant spatial information. There is a statistically significant correlation between probe accuracy and agent sample efficiency (steps to 80% success).

| Agent | Pool Size | Steps (M) | Linear Probe Accuracy (%) |
|---|---|---|---|
| PPO | 1 | $1.75_{\pm 0.44}$ | $99.81 \pm 0.14$ |
| | 5 | $7.80_{\pm 1.08}$ | $96.42 \pm 1.06$ |
| | 10 | $9.73_{\pm 0.36}$ | $87.84 \pm 1.40$ |
| | 20 | $10.00_{\pm 0.00}$ | $66.97 \pm 5.28$ |
| PPO+PT(ID) | 1 | $0.95_{\pm 0.21}$ | $99.81 \pm 0.12$ |
| | 5 | $5.55_{\pm 1.22}$ | $96.83 \pm 0.61$ |
| | 10 | $9.17_{\pm 1.10}$ | $89.54 \pm 0.95$ |
| PPO+PT(OOD) | 1 | $1.34_{\pm 0.42}$ | $99.59 \pm 0.33$ |
| | 5 | $7.03_{\pm 1.07}$ | $95.68 \pm 0.77$ |
| | 10 | $9.70_{\pm 0.41}$ | $88.90 \pm 1.11$ |
| SAC | 1 | $0.33_{\pm 0.07}$ | $100.00 \pm 0.00$ |
| | 5 | $0.91_{\pm 0.12}$ | $97.63 \pm 0.68$ |
| | 10 | $1.65_{\pm 0.31}$ | $93.34 \pm 0.48$ |
| | 20 | $4.52_{\pm 1.43}$ | $80.74 \pm 6.31$ |
| | 30 | $9.23_{\pm 0.96}$ | $66.69 \pm 8.41$ |
| | 50 | $10.00_{\pm 0.00}$ | $55.52 \pm 0.09$ |
| SAC+RAD | 1 | $0.24_{\pm 0.03}$ | $99.99 \pm 0.01$ |
| | 5 | $0.42_{\pm 0.06}$ | $98.66 \pm 0.20$ |
| | 10 | $0.82_{\pm 0.18}$ | $89.74 \pm 0.73$ |
| SAC+CURL | 1 | $0.46_{\pm 0.10}$ | $99.98 \pm 0.03$ |
| | 5 | $1.56_{\pm 0.31}$ | $97.14 \pm 0.17$ |
| | 10 | $5.24_{\pm 1.92}$ | $89.47 \pm 1.23$ |
| SAC+SPR | 1 | $2.09_{\pm 0.81}$ | $99.99 \pm 0.01$ |
| | 5 | $3.68_{\pm 1.68}$ | $94.31 \pm 0.24$ |
| | 10 | $10.00_{\pm 0.00}$ | $75.48 \pm 1.82$ |
| SAC+DBC | 1 | $0.44_{\pm 0.04}$ | $100.00 \pm 0.00$ |
| | 5 | $0.99_{\pm 0.25}$ | $94.26 \pm 1.19$ |
| | 10 | $10.00_{\pm 0.00}$ | $76.59 \pm 5.82$ |
| SAC+AE | 1 | $0.42_{\pm 0.09}$ | $100.00 \pm 0.00$ |
| | 5 | $1.04_{\pm 0.24}$ | $95.52 \pm 4.56$ |
| | 10 | $2.03_{\pm 0.38}$ | $88.66 \pm 1.88$ |
| SAC+VAE | 1 | $1.13_{\pm 0.14}$ | $99.66 \pm 0.06$ |
| | 5 | $5.30_{\pm 0.68}$ | $78.21 \pm 2.35$ |
| | 10 | $10.00_{\pm 0.00}$ | $64.76 \pm 0.11$ |
| SAC+SB | 1 | $0.98_{\pm 0.88}$ | $99.90 \pm 0.03$ |
| | 5 | $2.08_{\pm 0.30}$ | $96.69 \pm 1.08$ |
| | 10 | $10.00_{\pm 0.00}$ | $81.93 \pm 6.06$ |

C.2.1. ANALYSIS OF ALGORITHMIC VARIANTS

Our preliminary analysis of algorithmic variants suggests how method assumptions may influence effectiveness in SPGym. PPO with in-distribution pretraining (PT (ID)) significantly boosts sample efficiency across all pool sizes, though this advantage diminishes at pool size 10. Out-of-distribution pretraining (PT (OOD)) offers more modest gains that also decrease with larger pools, suggesting limited transfer from general-purpose pretrained encoders. For SAC, data augmentation via RAD consistently improves efficiency across all pool sizes, with particularly pronounced benefits for larger pools. Conversely, many sophisticated auxiliary methods struggle: CURL, SPR, and VAE variants consistently require more samples than standard SAC, with particularly poor performance on larger pools. DBC and AE generally underperform or offer marginal improvements. DreamerV3 demonstrates particularly strong performance, consistently outperforming both PPO and SAC variants across all pool sizes with remarkably stable performance. The variant without decoder gradients shows reduced performance, highlighting the importance of the reconstruction objective and suggesting that learning a predictive environment model provides an effective foundation for handling visual diversity. We now provide more detailed hypotheses for these behaviors.

**Pretraining.** As shown in Figure 10, pretraining provides clear benefits for PPO, especially with larger pools. While in-distribution pretraining provides strong gains, almost matching the performance of PPO with one-hot-based observations (see Appendix C.1.1), this represents an optimistic upper bound since real-world scenarios rarely permit task-specific pretraining. Interestingly, out-of-distribution pretraining also shows benefits compared to random initialization (90% vs 86% success at pool size 5), suggesting some transfer of useful visual features from the pretrained encoder. This indicates that even general-purpose pretrained encoders can provide a helpful initialization for RL tasks, though not matching the performance of task-specific pretraining.

**Data augmentation.** RAD succeeds by enforcing spatial invariances through grayscale+channel shuffling, preserving structural relationships critical for puzzle solving while adding beneficial stochasticity. Its weak assumptions make it robust across diversity levels, maintaining strong and consistent performance across pool sizes through this simple augmentation-based approach.

**Contrastive learning.** CURL underperforms as instance discrimination may prioritize whole-image features over tile-level details needed for puzzle solving. This suggests contrastive learning's focus on global image similarity may not align well with the local spatial reasoning required for puzzle solving.

**State similarity learning.** DBC fails possibly because its core assumption – that states with similar dynamics should have similar representations – breaks down in two ways: identical puzzle states appear radically distinct between episodes with different sampled images, while different states can share visual patterns due to being from the same episode or having the same base image.

**Temporal consistency.** While the environment's underlying dynamics are deterministic, temporal consistency methods such as SPR and SB face three key challenges: (1) The visual manifestation of state transitions varies dramatically between episodes due to different base images, forcing the encoder to learn position-invariant representations that capture tile relationships rather than visual content – a difficult disentanglement problem. (2) The assumption of smooth latent space transitions is violated by the discrete nature of tile movements, where single actions induce significant changes in both visual appearance and puzzle state. (3) Most crucially, these methods must simultaneously learn two competing objectives: temporal predictability in latent space (for transition modeling) and visual discriminability (for representation learning). This creates a conflict where features useful for predicting latent transitions (tile positions) are obscured by visually salient but dynamically irrelevant image content. SPR's prediction horizon mechanism exacerbates this by compounding representation errors through multiple latent transition steps. Similarly, SB's transition/reward prediction suffers because the latent space conflates visual features with positional information – while rewards depend solely on tile positions, the visual diversity in observations provides no direct positional cues. The same absolute position (e.g., top-left corner) shows completely different visual content each episode, making position-aware latent representations particularly difficult to learn.

**Reconstruction-based learning.** The success of DreamerV3's decoder highlights the value of the discrete reconstruction loss in learning useful representations for SPGym. In contrast, simple autoencoders (AE) offer little benefit for SAC, and variational autoencoders (VAE) hurt performance possibly because their continuous latent space assumptions conflict with SPGym's discrete state transitions.

These findings align with observations from Tomar et al. (2023), who noted that many representation learning methods underperform or fail completely when tested outside their original domain. We note that our evaluation focused on using each algorithm's suggested hyperparameters for visual RL with discrete actions, aiming to assess their out-of-the-box performance. The results may not reflect the best possible performance achievable through extensive hyperparameter tuning.

*Table 11.* **Performance metrics for PPO agents across image pool sizes.** 'Steps' indicates the number of environment steps (in millions) required to reach 80% success rate during training. 'ID Success' shows the final success rate at the end of 10M training steps when evaluating on the same pool used during training. 'OOD Easy' shows the success rate when evaluated on 100 unseen images from the 'Easy' distribution. 'OOD Hard' shows the success rate when evaluated on 100 unseen images from the 'Hard' distribution. 'Length' indicates the average number of steps required to solve an instance of the puzzle at the last 100 episodes.

| Algorithm | Variation | Pool | Steps (M) | ID Success (%) | OOD Easy (%) | OOD Hard (%) | Length |
|---|---|---|---|---|---|---|---|
| PPO | Standard | 1 | $1.75_{\pm0.44}$ | $100_{\pm0}$ | $0.49_{\pm0.13}$ | $0.0_{\pm0.0}$ | $72.9_{\pm4.4}$ |
| | | 5 | $7.80_{\pm1.08}$ | $86_{\pm16}$ | $0.53_{\pm0.14}$ | $0.3_{\pm0.3}$ | $212.0_{\pm19.0}$ |
| | | 10 | $9.73_{\pm0.36}$ | $47_{\pm18}$ | $0.34_{\pm0.08}$ | $0.6_{\pm0.4}$ | $599.6_{\pm25.6}$ |
| | | 20 | $10.00_{\pm0.00}$ | $12_{\pm7}$ | $0.12_{\pm0.03}$ | $0.0_{\pm0.0}$ | $903.4_{\pm23.9}$ |
| | PT (ID) | 1 | $0.95_{\pm0.21}$ | $100_{\pm0}$ | $0.33_{\pm0.09}$ | $0.1_{\pm0.1}$ | $76.2_{\pm4.8}$ |
| | | 5 | $5.55_{\pm1.22}$ | $100_{\pm0}$ | $0.53_{\pm0.16}$ | $0.3_{\pm0.4}$ | $83.2_{\pm7.2}$ |
| | | 10 | $9.17_{\pm1.10}$ | $38_{\pm20}$ | $0.27_{\pm0.07}$ | $0.9_{\pm0.7}$ | $658.4_{\pm25.3}$ |
| | PT (OOD) | 1 | $1.34_{\pm0.42}$ | $100_{\pm0}$ | $0.49_{\pm0.12}$ | $0.2_{\pm0.3}$ | $72.7_{\pm4.0}$ |
| | | 5 | $7.03_{\pm1.07}$ | $90_{\pm16}$ | $0.52_{\pm0.14}$ | $0.3_{\pm0.3}$ | $156.0_{\pm16.0}$ |
| | | 10 | $9.70_{\pm0.41}$ | $46_{\pm19}$ | $0.34_{\pm0.08}$ | $0.8_{\pm1.2}$ | $572.0_{\pm26.0}$ |

*Table 12.* **Performance metrics for SAC agents across image pool sizes.** 'Steps' indicates the number of environment steps (in millions) required to reach 80% success rate during training. 'ID Success' shows the final success rate at the end of 10M training steps when evaluating on the same pool used during training. 'OOD Easy' shows the success rate when evaluated on 100 unseen images from the 'Easy' distribution. 'OOD Hard' shows the success rate when evaluated on 100 unseen images from the 'Hard' distribution. 'Length' indicates the average number of steps required to solve an instance of the puzzle at the last 100 episodes.

| Algorithm | Variation | Pool | Steps (M) | ID Success (%) | OOD Easy (%) | OOD Hard (%) | Length |
|---|---|---|---|---|---|---|---|
| SAC | Standard | 1 | $0.33_{\pm0.07}$ | $100_{\pm0}$ | $0.45_{\pm0.12}$ | $0.0_{\pm0.0}$ | $86.7_{\pm19.5}$ |
| | | 5 | $0.91_{\pm0.12}$ | $100_{\pm0}$ | $0.58_{\pm0.12}$ | $0.0_{\pm0.0}$ | $76.8_{\pm16.2}$ |
| | | 10 | $1.65_{\pm0.31}$ | $100_{\pm0}$ | $0.46_{\pm0.12}$ | $0.0_{\pm0.0}$ | $78.7_{\pm16.5}$ |
| | | 20 | $4.52_{\pm1.43}$ | $98_{\pm2}$ | $0.35_{\pm0.11}$ | $0.0_{\pm0.0}$ | $63.2_{\pm18.4}$ |
| | | 30 | $9.23_{\pm0.96}$ | $47_{\pm38}$ | $0.19_{\pm0.04}$ | $0.0_{\pm0.0}$ | $525.0_{\pm63.1}$ |
| | | 50 | $10.00_{\pm0.00}$ | $7_{\pm5}$ | $0.06_{\pm0.02}$ | $0.0_{\pm0.0}$ | $917.0_{\pm35.5}$ |
| | RAD | 1 | $0.24_{\pm0.03}$ | $100_{\pm0}$ | $0.62_{\pm0.15}$ | $0.0_{\pm0.0}$ | $50.9_{\pm6.8}$ |
| | | 5 | $0.42_{\pm0.06}$ | $100_{\pm0}$ | $0.42_{\pm0.13}$ | $0.0_{\pm0.0}$ | $76.9_{\pm12.0}$ |
| | | 10 | $0.82_{\pm0.18}$ | $100_{\pm0}$ | $0.30_{\pm0.11}$ | $0.0_{\pm0.0}$ | $144.1_{\pm25.4}$ |
| | CURL | 1 | $0.46_{\pm0.10}$ | $100_{\pm0}$ | $0.76_{\pm0.09}$ | $0.0_{\pm0.0}$ | $77.2_{\pm13.1}$ |
| | | 5 | $1.56_{\pm0.31}$ | $100_{\pm0}$ | $0.44_{\pm0.10}$ | $0.2_{\pm0.4}$ | $84.9_{\pm17.7}$ |
| | | 10 | $5.24_{\pm1.92}$ | $100_{\pm0}$ | $0.37_{\pm0.11}$ | $0.0_{\pm0.0}$ | $88.0_{\pm20.3}$ |
| | SPR | 1 | $2.09_{\pm0.81}$ | $100_{\pm0}$ | $0.65_{\pm0.13}$ | $0.6_{\pm1.2}$ | $206.4_{\pm23.8}$ |
| | | 5 | $3.68_{\pm1.68}$ | $69_{\pm13}$ | $0.21_{\pm0.09}$ | $0.0_{\pm0.0}$ | $523.9_{\pm55.6}$ |
| | | 10 | $10.00_{\pm0.00}$ | $9_{\pm3}$ | $0.07_{\pm0.04}$ | $0.0_{\pm0.0}$ | $912.1_{\pm36.0}$ |
| | DBC | 1 | $0.44_{\pm0.04}$ | $100_{\pm0}$ | $0.44_{\pm0.13}$ | $0.0_{\pm0.0}$ | $65.1_{\pm13.8}$ |
| | | 5 | $0.99_{\pm0.25}$ | $100_{\pm0}$ | $0.34_{\pm0.13}$ | $0.0_{\pm0.0}$ | $111.2_{\pm22.7}$ |
| | | 10 | $10.00_{\pm0.00}$ | $2_{\pm1}$ | $0.13_{\pm0.04}$ | $0.0_{\pm0.0}$ | $588.5_{\pm63.3}$ |
| | AE | 1 | $0.42_{\pm0.09}$ | $100_{\pm0}$ | $0.78_{\pm0.11}$ | $0.0_{\pm0.0}$ | $85.4_{\pm22.1}$ |
| | | 5 | $1.04_{\pm0.24}$ | $100_{\pm0}$ | $0.64_{\pm0.16}$ | $0.0_{\pm0.0}$ | $102.6_{\pm22.6}$ |
| | | 10 | $2.03_{\pm0.38}$ | $100_{\pm0}$ | $0.55_{\pm0.12}$ | $1.3_{\pm2.6}$ | $78.0_{\pm17.1}$ |
| | VAE | 1 | $1.13_{\pm0.14}$ | $100_{\pm0}$ | $0.64_{\pm0.15}$ | $0.0_{\pm0.0}$ | $75.1_{\pm15.6}$ |
| | | 5 | $5.30_{\pm0.68}$ | $100_{\pm0}$ | $0.30_{\pm0.08}$ | $0.0_{\pm0.0}$ | $81.2_{\pm18.2}$ |
| | | 10 | $10.00_{\pm0.00}$ | $25_{\pm17}$ | $0.12_{\pm0.03}$ | $0.4_{\pm0.5}$ | $834.3_{\pm47.5}$ |
| | SB | 1 | $0.98_{\pm0.88}$ | $100_{\pm0}$ | $0.89_{\pm0.08}$ | $0.0_{\pm0.0}$ | $130.1_{\pm25.0}$ |
| | | 5 | $2.08_{\pm0.30}$ | $91_{\pm17}$ | $0.65_{\pm0.12}$ | $0.0_{\pm0.0}$ | $117.8_{\pm20.5}$ |
| | | 10 | $10.00_{\pm0.00}$ | $3_{\pm2}$ | $0.06_{\pm0.02}$ | $0.2_{\pm0.4}$ | $980.3_{\pm18.0}$ |

*Table 13.* **Performance metrics for DreamerV3 agents across image pool sizes.** 'Steps' indicates the number of environment steps (in millions) required to reach 80% success rate during training. 'ID Success' shows the final success rate at the end of 10M training steps when evaluating on the same pool used during training. 'Length' indicates the average number of steps required to solve an instance of the puzzle at the last 100 episodes.

| Algorithm | Variation | Pool | Steps (M) | ID Success (%) | Length |
|---|---|---|---|---|---|
| DreamerV3 | Standard | 1 | $0.42_{\pm0.06}$ | $100_{\pm0}$ | $83.5_{\pm11.8}$ |
| | | 5 | $1.23_{\pm0.20}$ | $100_{\pm0}$ | $31.0_{\pm3.9}$ |
| | | 10 | $1.44_{\pm0.58}$ | $100_{\pm0}$ | $32.9_{\pm4.0}$ |
| | | 20 | $3.96_{\pm0.61}$ | $100_{\pm0}$ | $27.8_{\pm3.5}$ |
| | | 30 | $5.84_{\pm0.71}$ | $99_{\pm1}$ | $38.0_{\pm6.2}$ |
| | | 50 | $6.62_{\pm2.67}$ | $87_{\pm14}$ | $177.8_{\pm24.8}$ |
| | | 100 | $8.18_{\pm2.33}$ | $29_{\pm14}$ | $676.1_{\pm32.8}$ |
| | w/o decoder | 1 | $1.13_{\pm0.12}$ | $100_{\pm0}$ | $36.0_{\pm2.5}$ |
| | | 5 | $1.79_{\pm0.61}$ | $100_{\pm0}$ | $42.1_{\pm3.2}$ |
| | | 10 | $2.57_{\pm0.91}$ | $100_{\pm0}$ | $46.7_{\pm3.9}$ |

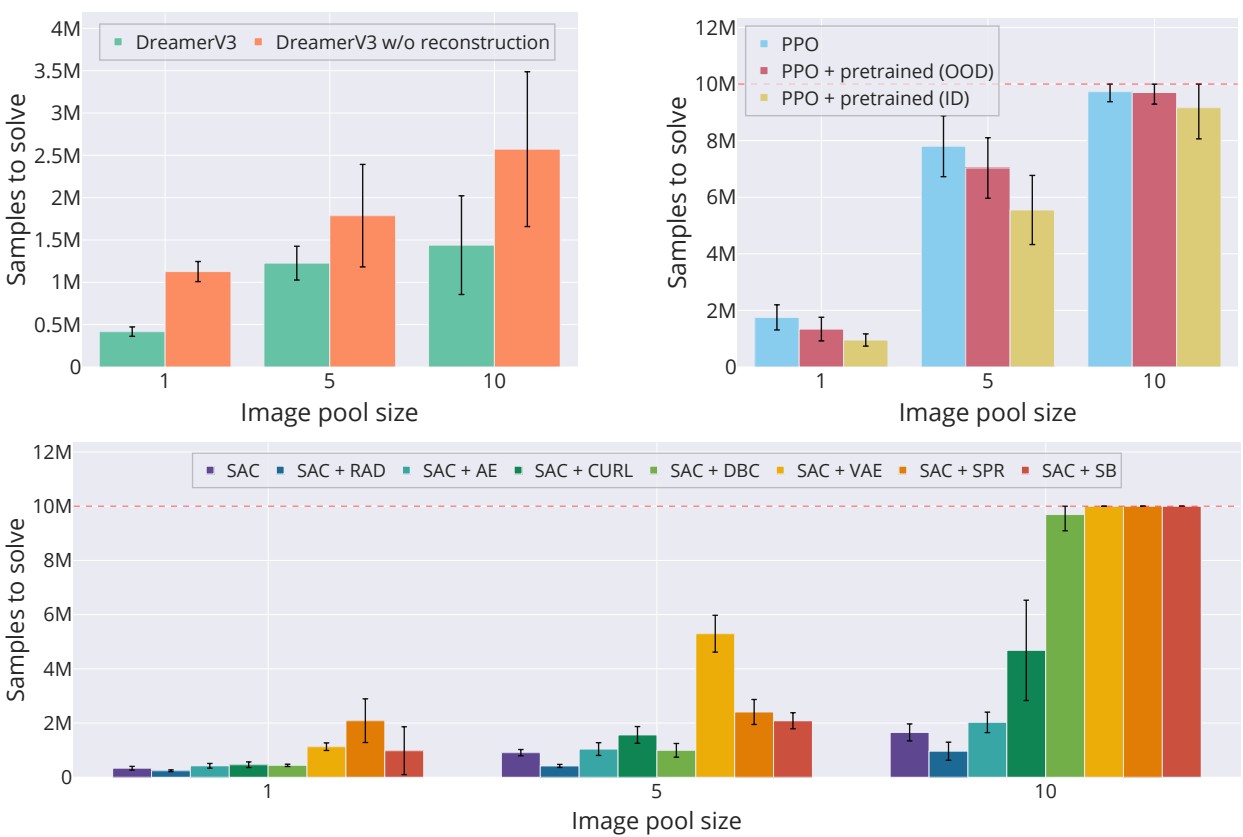

*Figure 9.* **Sample efficiency of different methods across pool sizes (lower is better).** SPGym differentiates agents based on their representation learning capabilities. Top left: DreamerV3 variants demonstrate the value of reconstruction learning. Top right: PPO results show benefits of pretraining. Bottom: Comprehensive comparison of SAC variants reveals trade-offs between different representation learning approaches.

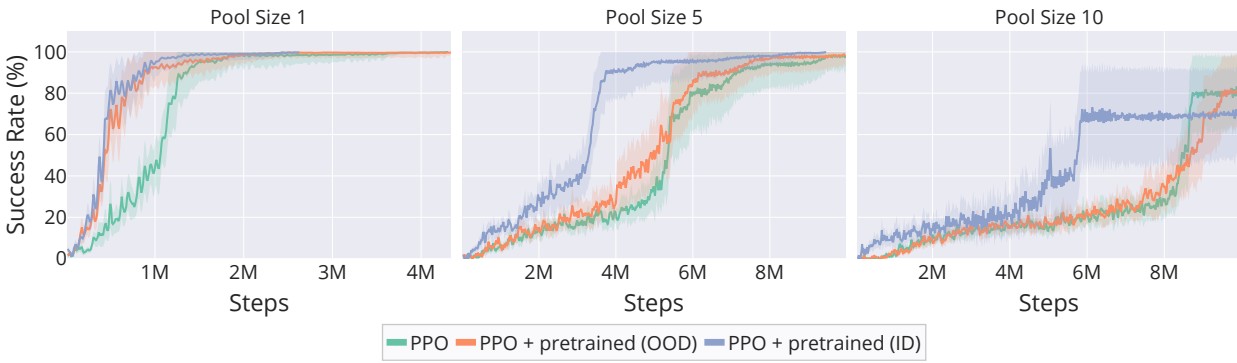

*Figure 10.* **Learning curves for PPO variants.** Success rate during training for baseline PPO and versions with pretrained encoders across different pool sizes. Shaded regions represent 95% confidence intervals across 5 independent runs.

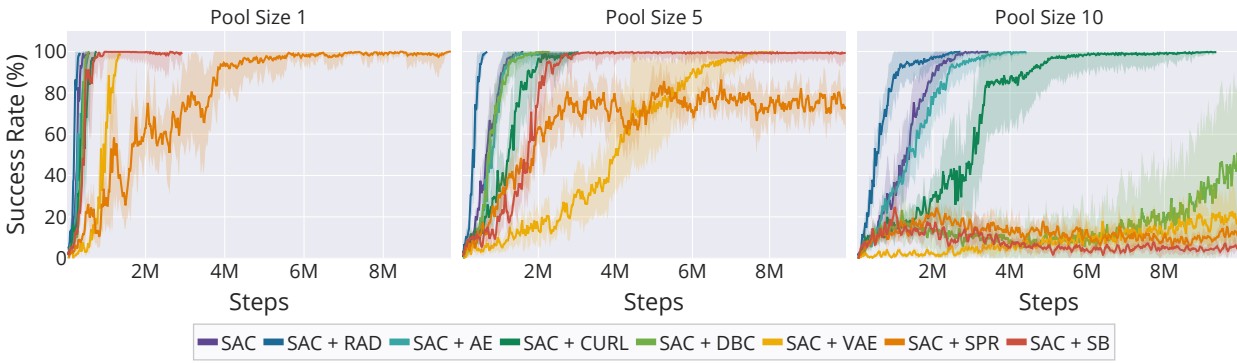

*Figure 11.* **Learning curves for SAC variants.** Success rate during training for baseline SAC and versions with different representation learning components across different pool sizes. Shaded regions represent 95% confidence intervals across 5 independent runs.

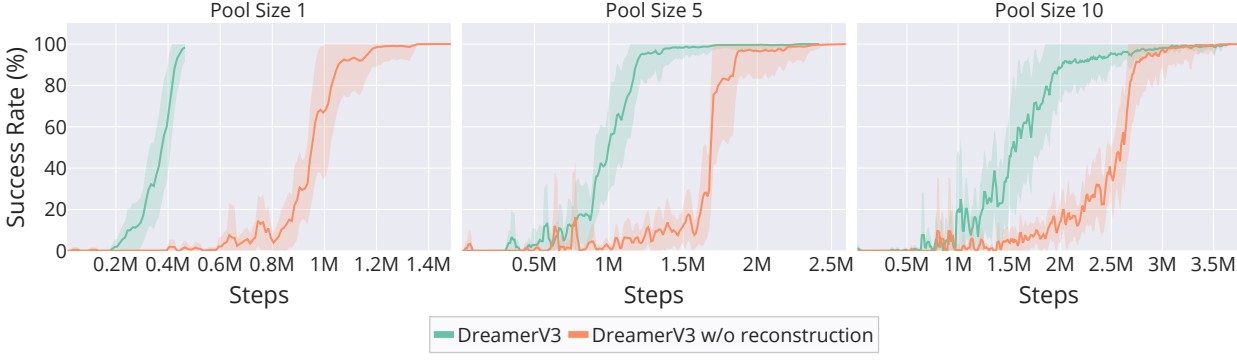

*Figure 12.* **Learning curves for DreamerV3 variants.** Success rate during training for DreamerV3 with and without decoder across different pool sizes. Shaded regions represent 95% confidence intervals across 5 independent runs.

