# OpenReview forum: "Sliding Puzzles Gym: A Scalable Benchmark for State Representation in Visual Reinforcement Learning"
_ICML.cc/2025/Conference — ICML 2025 poster_

### Official Review · Reviewer_R6Sn · 2025-03-10

**Overall Recommendation:** 3

**Summary:**

The paper introduces a sliding puzzle based environment for evaluating visual RL. It provides a number of baselines on the environment.

**Claims And Evidence:**

Yes.

**Essential References Not Discussed:**

N/A

**Experimental Designs Or Analyses:**

Yes. It is sound.

**Methods And Evaluation Criteria:**

Yes.

**Other Comments Or Suggestions:**

N/A

**Other Strengths And Weaknesses:**

Weakness related to writing: it is still not clear to me why this method would test an RL agent's visual representation capability? Clearly, an RL agent needs to put together _what_ is the goal image. It only understands so when the goal is an image that _makes sense_. From that point of view, it is not convincing to if we are _only_ evaluating the visual aspect of RL.

I would encourage authors to provide a better motivation to this question in the paper. Based on the response, I am open to changing my evaluation (both positive and negative).

**Questions For Authors:**

Please see above.

**Relation To Broader Scientific Literature:**

The authors do a good job discussing the previous works such as distracting DMC, vanilla DMC, ProcGen, etc.

**Theoretical Claims:**

N/A

---

> ### Author Rebuttal · Authors · 2025-04-01
>
> We thank the reviewer for their positive assessment of our experimental design and literature review, and for the opportunity to clarify the core motivation behind SPGym regarding the evaluation of visual representation capabilities.
>
> ## Why SPGym Tests Visual Representation Capabilities
>
> The central idea behind SPGym is to create a controlled environment where the primary challenge being scaled is the agent's ability to process and understand diverse visual inputs, even though the agent is trained end-to-end on a standard RL task. We achieve this through specific design choices:
>
> 1.  **POMDP Formulation:** As detailed in Section 3, SPGym is formulated as a Partially Observable Markov Decision Process (POMDP) defined by $(\mathcal{S}, \mathcal{X}, \mathcal{A}, \mathcal{P}, \mathcal{R}, \mathcal{S}_0)$. Critically, the agent never observes the true underlying puzzle state $s \in \mathcal{S}$. Instead, it only receives visual observations $x \in \mathcal{X}$. The agent must infer the relevant state information solely from these high-dimensional visual inputs.
>
> 2.  **Isolating the Visual Challenge:** The key design element is that across all experiments and all compared agents, the core components of the MDP remain fixed: the state space $\mathcal{S}$ (tile permutations), the action space $\mathcal{A}$ (up, down, left, right), the deterministic transition dynamics $\mathcal{P}$ (how tiles move), the reward function $\mathcal{R}$ (based on Manhattan distance), and the initial state distribution $\mathcal{S}_0$. The only thing that changes between experimental conditions (e.g., different pool sizes) or agent comparisons (e.g., different representation learning modules) is the emission function that maps the underlying state $s$ to the visual observation $x$. We vary this by overlaying the puzzle state with different pools of images.
>
> 3.  **Performance Reflects Representation Quality:** Because all core task elements (dynamics, rewards, etc.) are constant, any difference in performance (e.g., sample efficiency) between agents or across different visual diversity levels must be attributed to how effectively the agent's visual encoder maps the observation $x$ to a useful internal representation. An agent cannot solve the puzzle without implicitly or explicitly understanding the tile configuration from the image. Better visual representations enable the policy to make better decisions, leading to faster learning and higher success rates. While policy learning is intrinsically linked, the bottleneck being systematically stressed and evaluated is the visual representation learning component.
>
> ### Empirical Evidence
>
> To provide further empirical support for this link between representation quality and task performance, we conducted linear probe evaluations (inspired by feedback from reviewer o6ZL) on the frozen encoders learned by PPO and SAC agents. We trained linear classifiers to predict tile positions from the learned features. Our results show a statistically significant negative correlation (Pearson r=-0.81, p=1.1e-13) between the probe's test accuracy and the number of environment steps the RL agent needed to reach 80% success. This demonstrates that encoders capable of producing more informative features (higher probe accuracy) enable faster learning on the downstream RL task. The probe accuracies for tested agents follows:
>
> |Agent/Pool|1|5|10|20|30|50|
> |-|-|-|-|-|-|-|
> |PPO|99.81±0.14|96.42±1.06|87.84±1.40|66.97±5.28|-||
> |PPO+PT(ID)|99.81±0.12|96.83±0.61|89.54±0.95|-||-|
> |PPO+PT(OOD)|99.59±0.33|95.68±0.77|88.90±1.11|-||-|
> |SAC|100.00±0.00|97.63±0.68|93.34±0.48|80.74±6.31|66.69±8.41|55.52±0.09|
> |SAC+RAD|99.99±0.01|98.66±0.20|89.74±0.73|-||-|
> |SAC+CURL|99.98±0.03|97.14±0.17|89.47±1.23|-||-|
> |SAC+SPR|99.99±0.01|94.31±0.24|75.48±1.82|-||-|
> |SAC+DBC|100.00±0.00|94.26±1.19|76.59±5.82|-||-|
> |SAC+AE|100.00±0.00|95.52±4.56|88.66±1.88|-||-|
> |SAC+VAE|99.66±0.06|78.21±2.35|64.76±0.11|-||-|
> |SAC+SB|99.90±0.03|96.69±1.08|81.93±6.06|-||-|
>
> ## Conclusion
>
> In essence, while the agent learns end-to-end, SPGym isolates the difficulty scaling to the visual domain. An agent succeeding in SPGym, especially across varying image pools, demonstrates robustness in its visual processing specific to the task structure. We believe this setup effectively probes an agent's ability to form useful representations from pixels under varying visual conditions, which is a critical aspect of visual RL.
>
> We will revise the Introduction and Methodology sections to incorporate this detailed explanation and motivation, ensuring it is clear how our design choices allow for the evaluation of visual representation learning capabilities in a controlled manner. We hope this addresses the reviewer's concern and clarifies the value proposition of SPGym. We appreciate the reviewer's willingness to reconsider their evaluation based on this clarification.

---

> > ### Comment · Reviewer_R6Sn · 2025-04-04
> >
> > While I appreciate more experiments in such short time, I am unsure the results shown here answers my question and remain unconvinced. I _do_ think the claim is okay (hence leaning accept), except that the current experiments do a _subpar_ job of showing it.
> >
> > I suggest a simple experimental setup: can we have a ground truth latent for the environment observations? Can an RL agent learn significantly better for a large number of different environments given the ground truth latent?
> >
> > I will stay at my borderline accept rating and happy to go up if the authors can present a convincing experiment.

---

> > > ### Author Response · Authors · 2025-04-09
> > >
> > > Thank you for the further discussion and the constructive suggestion for a direct comparison experiment. We agree this is a valuable way to assess the impact of learning from visual observations versus ground-truth states.
> > >
> > > Following your suggestion, we trained PPO, SAC, and DreamerV3 agents using SPGym's one-hot encoding variation (representing the ground-truth puzzle state, identical to the targets for our linear probes) and compared their sample efficiency (steps to 80% success, avg. 5 seeds) against the image-based versions. For PPO and SAC, we replaced the CNN encoders with 2-layer MLPs to process the one-hot vectors. DreamerV3 used its default non-image encoder (a 3-layer MLP). We maintained hyperparameters close to the image-based experiments without specific tuning for the one-hot setting.
> > >
> > > |Algorithm|Grid Size|One-hot|Image (Pool 1)|Image (Pool 5)|
> > > |-|-|-|-|-|
> > > |PPO|3x3|661.69k±81.44k|1.75M±444.81k|7.80M±1.08M|
> > > ||4x4|12.29M±467.84k|24.46M±7.58M|-|
> > > |SAC|3x3|672.51k±63.10k|334.26k±67.47k|907.21k±116.20k|
> > > ||4x4|5.09M±463.14k|8.14M±3.64M|-|
> > > |DreamerV3|3x3|834.86k±61.10k|417.09k±55.03k|1.23M±199.49k|
> > > ||4x4|3.68M±436.97k|2.26M±287.23k|5.81M ± 2.17M|
> > >
> > > These results provide several insights. For PPO on both grid sizes and SAC on the 4x4 grid, learning directly from the ground-truth one-hot state is more sample efficient than learning from images. The results for SAC and DreamerV3 on the 3x3 grid, where pool 1 images led to faster convergence than one-hot, may be influenced by the differences in network architectures and the lack of architecture/hyperparameter tuning specifically for the one-hot setting.
> > >
> > > Crucially, however, across all agents and grid sizes, increasing the visual diversity from **image pool size 1 to pool size 5 and beyond consistently increases the sample complexity**. This shows the impact of the visual representation challenge that SPGym is designed to probe, isolating the effect of visual diversity on learning efficiency.
> > >
> > > While the one-hot version provides a useful ground-truth baseline, **its difficulty is fixed**. SPGym's core value lies in its image-based variations, which allow us to systematically scale the visual diversity challenge (Pool 1 vs. Pool 5 vs. Pool 10, etc., see Table 5 in the main paper) while keeping the underlying task dynamics constant. This enables the controlled evaluation of how effectively different RL agents learn representations under this specific, scalable stress, revealing limitations that wouldn't be apparent from the one-hot setting alone. While acknowledging that perfect disentanglement is challenging in end-to-end learning, SPGym provides a framework for this structured, comparative evaluation of visual representation learning capabilities in RL.
> > >
> > > We thank you again for pushing us on this and for providing the suggestion that led to these insightful results. We will incorporate this experiment and discussion into the revised manuscript.

---

### Official Review · Reviewer_ukCk · 2025-03-14

**Overall Recommendation:** 3

**Summary:**

This paper presents SPGym, a new benchmark for visual reinforcement learning (RL) based on the classic 8-tile puzzle. SPGym uses a visual observation space derived from large datasets and allows researchers to manipulate representation complexity by adjusting visual diversity. Experiments using model-free and model-based RL algorithms on SPGym reveal that current methods struggle to generalize across varying visual inputs, with performance degrading as visual diversity increases.

**Claims And Evidence:**

The reviewer agrees that the ability to "scale the representation learning complexity while maintaining consistent environment dynamics" is a crucial aspect of research in visual reinforcement learning. The proposed SPGym environment, despite its simplicity, demonstrates potential in fulfilling this objective. By merely utilizing different images within the puzzle, one can effectively control the visual complexity of the task.

**Essential References Not Discussed:**

None

**Experimental Designs Or Analyses:**

In addition to the issues discussed in 'Methods And Evaluation Criteria' section, the reviewer has concerns regarding the size of the training sets. Utilizing only five images in the training set appears insufficient, and the reviewer finds it unsurprising that existing algorithms fail to generalize to unseen images under these conditions. While the authors mention experimenting with training sets containing up to 100 images, this quantity is still considered limited by the reviewer. The reviewer suggests exploring the use of significantly larger training sets, on the order of 100,000 or even 1 million images, to assess the impact on generalization performance.

**Methods And Evaluation Criteria:**

The reviewer has concerns regarding the evaluation settings.

1. In the 'in distribution' setting, there appears to be no separate hold-out validation set for evaluating the learned policy. Given the deterministic nature of the game, the reported numbers in Table 2 seem to only reflect how well each method overfits to the five training game instances. The reviewer found that the results did not provide clear conclusions or insights about the tested algorithms.

2. Table 3 presents results that suggest none of the existing algorithms were able to solve the puzzle when the testing image was not included in the training pool. This indicates a significant limitation of the benchmark. A benchmark that results in a zero success rate for all RL algorithms does not provide a useful basis for comparison or meaningful insights into the relative strengths and weaknesses of different algorithms.

The reviewer believes that the SPGym environment still holds potential despite the aforementioned issues.

For instance, one could use the similarity between training and testing images as a parameter to control the difficulty of the visual task. This could involve a tiered approach:

- Easy Level: The testing images could be augmented versions of the training images.
- Intermediate Level: The testing images could be drawn from the same classes as the training images.
- Hard Level: The testing images could be sourced from entirely different datasets than the training images.

By evaluating different RL algorithms across these varying levels of visual difficulty, researchers could gain deeper insights into the visual capabilities and generalization abilities of these algorithms.

**Other Comments Or Suggestions:**

None

**Other Strengths And Weaknesses:**

The paper is generally easy to follow and well organized.

**Questions For Authors:**

Please see the above discussions.

**Relation To Broader Scientific Literature:**

This paper has close connections to visual reinforcement learning, such as [a, b]. The goal of the paper is to create an environment that can separate visual complexity from dynamics, which is indeed an important topics in the field.

[a] Revisiting Plasticity in Visual Reinforcement Learning: Data, Modules and Training Stages, Ma, 2024.

[b] DrM: Mastering Visual Reinforcement Learning through Dormant Ratio Minimization, Xu, 2024.

**Theoretical Claims:**

No formal theoretical claim is presented.

---

> ### Author Rebuttal · Authors · 2025-04-01
>
> We thank the reviewer for the positive feedback on SPGym's potential and the detailed, constructive comments regarding the evaluation settings and experimental design. We address the concerns below:
>
> ## 1. In-Distribution Evaluation
>
> We understand the concern that Table 2 results might seem like overfitting. However, our primary goal with the in-distribution (ID) setting is not to evaluate generalization across images but to measure the sample efficiency with which different RL agents learn useful visual representations under controlled visual diversity. We chose pool size 5 because it offers a balance: it introduces enough visual diversity to discriminate between different representation learning approaches (as seen by the varying steps-to-convergence) while remaining learnable for most agents within our computational budget.
>
> As detailed in our response to Reviewer R6Sn, SPGym is formulated as a POMDP where only the visual observation function changes between settings; the underlying task dynamics remain fixed. Therefore, differences in sample efficiency (Table 2) directly reflect how effectively each agent's representation learning component handles the visual aspect of the task.
>
> ## 2. Out-of-Distribution Evaluation
>
> We acknowledge the reviewer's point that zero OOD success might seem like a limitation. However, we view this consistent failure not as a flaw of the benchmark, but as a crucial diagnostic finding about the limitations of current end-to-end visual RL methods when faced with generalizing visual features learned purely through a task-specific RL objective. Standard RL objectives, like the one used here, do not explicitly optimize for generalization to unseen visual appearances of the same underlying state. SPGym effectively highlights this gap.
>
> Our choice to present the main OOD results (Table 3) using agents trained on pool size 5 stems from this being the setting where most methods achieved high ID success. Evaluating OOD generalization is most meaningful when agents have demonstrated competence on their training distribution. While some base agents achieved ID success on larger pools, they still failed completely OOD, reinforcing the finding that simply increasing training diversity within this range wasn't sufficient for generalization with current methods. Showing zero OOD success across methods aims to motivate research into algorithms with better visual generalization properties.
>
> ## 3. Training Set Size
>
> We appreciate the suggestion to explore significantly larger training pools. In preliminary experiments, we attempted training with pools of thousands of images. However, we found that with such high visual diversity, the RL training became unstable. Because each observation was essentially unique (rarely seen more than once), the gradients from the RL objective were insufficient to train a useful visual encoder from scratch. Without a stable encoder, the policy failed to learn.
>
> Therefore, our approach was to identify the approximate limit of visual diversity that current standard algorithms (PPO, SAC, DreamerV3) could handle within a reasonable budget (10M steps), which led to testing pools up to 20, 50, and 100, respectively. This reveals the scaling limitations of these methods rather than aiming for OOD generalization via massive datasets, which might require different training paradigms (e.g., pretraining on external data, different objectives).
>
> ## 4. Tiered Generalization Evaluation
>
> We thank the reviewer for this excellent suggestion. We agree that evaluating generalization across different levels of visual similarity would provide deeper insights. As a preliminary step in this direction, we evaluated the trained PPO and SAC agents on augmented versions of their training images. We observed a strong correlation (Pearson r=-0.81, p=2.5e-12) between success rates on these augmented images and the agents' sample efficiency, suggesting a link between robustness to simple transformations and learning speed. We plan to include these results and discuss the tiered approach as a key direction for future work in the revised manuscript:
>
> |Agent/Pool|1|5|10|20|30|50|
> |-|-|-|-|-|-|-|
> |PPO|0.49±0.13|0.53±0.14|0.34±0.08|0.12±0.03|-|-|
> |PPO+PT(ID)|0.33±0.09|0.53±0.16|0.27±0.07|-|-|-|
> |PPO+PT(OOD)|0.49±0.12|0.52±0.14|0.34±0.08|-|-|-|
> |SAC|0.45±0.12|0.58±0.12|0.46±0.12|0.35±0.11|0.19±0.04|0.06±0.02|
> |SAC+AE|0.78±0.11|0.64±0.16|0.55±0.12|-|-|-|
> |SAC+VAE|0.64±0.15|0.30±0.08|0.12±0.03|-|-|-|
> |SAC+SPR|0.65±0.13|0.21±0.09|0.07±0.04|-|-|-|
> |SAC+DBC|0.44±0.13|0.34±0.13|0.13±0.04|-|-|-|
> |SAC+CURL|0.76±0.09|0.44±0.10|0.37±0.11|-|-|-|
> |SAC+RAD|0.62±0.15|0.42±0.13|0.30±0.11|-|-|-|
> |SAC+SB|0.89±0.08|0.65±0.12|0.06±0.02|-|-|-|
>
> We will revise the paper to clarify the evaluation rationale, incorporate the results on augmented images, and explicitly frame the tiered generalization as important future work, addressing the points raised. Thank you again for the constructive feedback.

---

### Official Review · Reviewer_o6ZL · 2025-03-17

**Overall Recommendation:** 3

**Summary:**

The paper introduces SPGym, a novel benchmark for visual RL that extends the classic sliding puzzle by replacing numbered tiles with image patches. This enables scaling visual diversity while keeping the puzzle dynamics fixed, with the aim of isolating representation learning from policy learning. The authors evaluate a range of RL baselines: on-policy PPO (with ID and OOD pretrained encoders), off-policy SAC with data augmentation and representation learning techniques, and the model-based DreamerV3, to assess sample efficiency and generalization. Their experiments show that pretraining and data augmentation enhance sample efficiency, baselines react differently to increasing the images in the training pool, nearly all approaches achieve high in-distribution performance yet completely fail to generalize to unseen images, and PPO continues to learn more efficient solutions after reaching 100% average success.

**Claims And Evidence:**

1. The authors repeatedly claim that SPGym disentangles representation learning from policy learning, using this as a core motivation for their work and distinguishing it from prior benchmarks like The Distracting Control Suite and ProcGen. However, I don’t see how their experimental setup fully supports this claim. While PPO is tested with pretrained encoders (both in-distribution and out-of-distribution), SAC and DreamerV3 still learn representations end-to-end, meaning representation learning is not truly isolated across all agents. SPGym does not incorporate explicit representation learning evaluations, such as frozen encoder tests or linear probes, which would help validate the claim of disentanglement. Since policy updates inherently shape learned representations, the benchmark does not provide a clear separation between the two processes.
2. The authors claim that the best-performing agents fall short of the theoretical 22-step optimal solution, basing this assertion on the 5-pool training results in Table 2. However, agents like SAC and DreamerV3 benefit from larger training pools, with DreamerV3 achieving the lowest episode length of 27.8 steps using a 20-image pool (Table 5). Moreover, since early stopping is applied, preventing further improvements, there is evidence from PPO that continued training can significantly reduce episode lengths (from 214.30 to 31.35 steps). Given that the best version of DreamerV3 is trained using only about 40% of the allowed data budget, it has the potential to approach the theoretical optimal solution with extended training. Therefore, the claim that the best-performing agents fall short of the optimal solution is not fully substantiated.

**Essential References Not Discussed:**

There are other benchmarks tailored for visual RL that have a setting which keeps the environment dynamics the same while increasing visual diversity [1, 2], similar to SPGym.

[1] Dosovitskiy, Alexey, et al. "CARLA: An open urban driving simulator." *Conference on robot learning*. PMLR, 2017.

[2] Tomilin, Tristan, et al. "Coom: A game benchmark for continual reinforcement learning." *Advances in Neural Information Processing Systems* 36 (2023): 67794-67832.

**Experimental Designs Or Analyses:**

1. The authors evaluate a diverse set of SOTA baselines, including on-policy PPO with pretrained encoders, off-policy SAC with various data augmentation strategies, and multiple recent variants from the literature, along with the model-based DreamerV3. This strengthens the paper’s contributions.
2. The baseline comparison of increasing the training pool size is insightful, as it reveals how well the algorithms adapt to greater visual diversity.
3. In Table 2, the authors assess sample efficiency by measuring the number of samples required for baseline methods to reach an 80% success rate with a small image pool. Establishing a clear evaluation criterion for sample efficiency is beneficial for the benchmark. However, while they mention that early stopping occurs when the agent achieves a 100% success rate for 100 consecutive episodes, they do not specify the window size used for averaging the 80% success rate threshold. This omission makes it unclear how success rates are computed and whether short-term fluctuations could affect the reported results.
4. The authors determine the best data augmentation strategy by evaluating only **RAD**, then apply the selected approach (grayscale + channel shuffle) to all augmentation-based methods, including **CURL** and **SPR**. This is problematic because different methods have distinct augmentation requirements. **CURL** [1] has been shown to benefit more from cropping rather than random color shuffling, while **SPR** [2] relies on temporal consistency and may be particularly sensitive to spatial distortions. Optimizing augmentations solely for RAD does not guarantee optimal performance for other methods, potentially skewing the results and disadvantaging CURL and SPR. This could explain why RAD outperforms CURL and SPR in sample efficiency.
5. The Hyperparameter Selection and Data Augmentation Analysis focus solely on optimizing training performance. However, since the benchmark also assesses evaluation, the selection process should consider generalization performance, not just faster learning. The chosen hyperparameters may improve training efficiency but contribute to overfitting, making it crucial to evaluate augmentation methods based on their impact on generalization.
6. Table 2 shows that pretrained encoders improve PPO’s sample efficiency. However, this comparison is not entirely fair, as the pretrained encoders have already seen a large number of samples during pretraining, even if from OOD data. This gives them a significant advantage over agents learning representations from scratch, making direct comparisons of sample efficiency misleading.
7. Table 3 presents the ID and OOD evaluation results, highlighting SPGym’s effectiveness as a generalization evaluation tool. While a training pool of only 5 images is understandably insufficient for meaningful generalization, the authors note that even models trained on pools of up to 100 images fail to transfer to unseen images. In my view, this shows the benchmark’s core challenge, demonstrating that overfitting remains a problem regardless of pool size. However, since SAC and DreamerV3 achieve strong in-distribution performance with larger training pools over 10M steps, it would be more informative to display their main results in the table on a larger pool size to better assess their generalization potential. Additionally, the authors only evaluate OOD performance for the base PPO, SAC, and DreamerV3 models. Extending this analysis to other baselines included in the study, such as data-augmented or contrastive learning variants, would provide a more comprehensive understanding of how different representation learning methods handle distribution shifts.
8. The authors apply early stopping once agents achieve a 100% success rate for 100 consecutive episodes, yet their own results show that agents continue to improve solution efficiency well beyond this point. The first 100 successful episodes average 214.30 steps, whereas the final 100 episodes average just 31.35 steps, indicating that further training yields significantly more optimal solutions. This raises concerns about the motivation behind early stopping in this setting. Typically, early stopping is used when a model has converged and is no longer learning anything meaningful. Here, it appears to halt training prematurely, limiting the evaluation of efficiency improvements.
9. The use of procedurally generated images, such as those from DiffusionDB, offers advantages like reduced storage overhead and access to near-infinite diversity, potentially aiding controlled generalization. However, Figure 5 shows that performance trends closely mirror those on ImageNet, suggesting that this diversity does not significantly impact learning outcomes under the tested conditions. While procedural generation allows fine-grained control over visual complexity, the results do not demonstrate a clear advantage over large pre-existing datasets. To strengthen this contribution, the authors could analyze its benefits in terms of computational efficiency, storage, or controlled generalization.

[1] Laskin, Michael, Aravind Srinivas, and Pieter Abbeel. "Curl: Contrastive unsupervised representations for reinforcement learning." *International conference on machine learning*. PMLR, 2020.

[2] Schwarzer, Max, et al. "Data-efficient reinforcement learning with self-predictive representations." *arXiv preprint arXiv:2007.05929* (2020).

**Methods And Evaluation Criteria:**

1. SPGym has two clearly defined training evaluation metrics: 1) **task completion**, measuring whether an agent successfully solves the puzzle within the maximum episode length of 1,000 steps, and 2) **completion efficiency**, measuring how quickly an agent solves the puzzle in terms of the number of steps taken. These well-defined metrics make evaluation and comparison straightforward.
2. SPGym measures sample efficiency by tracking the number of environment steps an agent requires to achieve an 80% success rate.
3. Increasing the training image pool size introduces greater visual diversity while keeping the underlying dynamics unchanged, allowing for an analysis of how different algorithms perform across varying levels of visual complexity (Figure 4 & Table 5).
4. SPGym is effective for evaluating generalization to unseen data, as all baseline methods fail to transfer their learned representations to unseen images, regardless of the training pool size.
5. One of my main concerns is the **reward function**. Using the Manhattan distance between each tile’s current and target positions as a reward signal discourages moves that might be necessary for reaching the optimal solution. There are likely scenarios where a tile that is already in place, or close to it, must be temporarily moved further away to solve the puzzle efficiently. This reward function risks biasing the agent toward locally optimal but globally suboptimal strategies, which could explain the poor baseline performance. While I acknowledge that a purely sparse +1 reward upon completion would make learning infeasible due to the extremely low probability of solving even a 3×3 puzzle through random actions, the current approach may still be counterproductive. The authors should design a dense reward signal that does not penalize necessary intermediate steps or provide empirical evidence that the current formulation does not hinder learning.

**Other Comments Or Suggestions:**

1. The authors provide wrappers that allow for different observation modalities beyond image-based tiles, including text and one-hot encodings. While this adds flexibility to SPGym, it is not a fully realized contribution since there is no empirical analysis or baseline evaluation using these alternative modalities. Without an evaluation of how different modalities impact learning performance, this aspect remains more of a hypothetical feature rather than a demonstrated advantage.
2. The authors state that they explore three algorithmic paradigms: off-policy, on-policy, and model-based. However, model-based methods can themselves be either off-policy or on-policy, depending on how data is used. The distinction could be better clarified for conceptual accuracy.
3. The evaluation of data augmentation methods is solely based on their impact on training performance, without assessing their effect on generalization during evaluation.
4. It would be interesting to analyze whether some images make it more difficult to solve the puzzle than others.
5. The authors repeatedly state that increasing the training pool size increases visual complexity. However, I believe it only increases visual diversity rather than complexity, unless certain images make it more difficult for the agent to solve the puzzle.

**Other Strengths And Weaknesses:**

1. Since the authors introduce a new benchmark, access to the code would be beneficial for properly assessing their work. However, they have not uploaded their code or provided a link to an anonymous repository.
2. This benchmark has strong potential to be valuable to the community. The 8-Puzzle problem with visual observations is a commendable proposal, particularly since it aligns with how humans approach the task. Benchmarks introducing novel problems are always beneficial. However, I am not fully convinced by the strong emphasis on disentangling feature and policy learning, as the current evidence does not fully support this claim. Additionally, some aspects of the experimental evaluation need improvement, as outlined above. I am very open to raising my score if the authors adequately address my concerns.

**Questions For Authors:**

1. Is the missing tile always in the bottom-right corner, or is it random across episodes?
2. Why do the authors use early stopping if the model still has the potential to learn more efficient solutions?
3. Does PPO converge when trained for 10M steps on the 4x4 grid, or could longer training improve performance further?

**Relation To Broader Scientific Literature:**

The authors position their work among benchmarks that introduce visual diversity, such as ProcGen and DM Control Suite, and those that add visual noise, like the Distracting Control Suite. They claim that SPGym uniquely disentangles representation learning from policy optimization by extending the 8-Puzzle task, previously used in RL research, to support visual observations.

**Theoretical Claims:**

This paper does not contain theoretical claims or formal proofs; its primary contributions are experimental and methodological.

---

> ### Author Rebuttal · Authors · 2025-04-01
>
> We sincerely thank the reviewer for their thorough and constructive feedback. We appreciate the recognition of SPGym's potential and the detailed suggestions, which will significantly improve the paper.
>
> ## 1. Answers to Direct Questions
>
> 1.  **Missing Tile:** The missing tile's starting position is randomized in each episode as part of the initial shuffling.
> 2.  **Training Termination ("Early Stopping"):** Our primary motivation for terminating runs based on sustained success rate was computational efficiency, given the number of experiments. A secondary reason was to evaluate OOD generalization before potential extreme overfitting. We agree "early stopping" is imprecise terminology, as agents can still improve solution efficiency, and will rephrase this in the revision.
> 3.  **PPO 4x4 Convergence:** PPO does not converge within 10M steps on the 4x4 grid. Extended runs (100M steps) show it requires approximately 24.5M ± 7.6M steps (avg. 5 seeds) to reach 80% success, indicating feasibility but requiring much more data. We will add this finding.
>
> ## 2. Addressing Key Concerns
>
> 1.  **Disentanglement of Representation and Policy Learning:** We acknowledge the reviewer's point on end-to-end learning. However, SPGym's design isolates the visual representation challenge as the key variable by keeping all underlying MDP components fixed and only varying the visual observation function. Performance differences thus reflect agents' visual processing capabilities. For a detailed explanation of the POMDP formulation and supporting empirical evidence from linear probing, please see our response to **Reviewer R6Sn**. We will add these clarifications and results to the revision.
>
> 2.  **Optimal Solution Claim:** You are correct. Given our termination criterion and that only PPO was tested for continued training, the claim that best-performing agents fall short might be too strong. We will revise this to reflect that agents under our protocol didn't reach the optimum, acknowledging the potential for improvement with longer training.
>
> 3.  **Reward Function:** The Manhattan distance reward is standard in puzzle literature (Korf et al., 1985; Burns et al., 2012; Lee et al., 2022; Moon et al., 2024) and encourages minimizing steps by accumulating negative rewards until the goal (+1) is reached. While local optima are possible, this aligns the RL objective with finding efficient solutions. A perfect dense reward is non-trivial, and sparse rewards are infeasible here.
>
> 4.  **Evaluation Criteria Clarity (80% Threshold):** We apologize for the omission. We calculate this by finding the first environment step where the average success rate across all parallel environments reaches 80% for each run, then averaging these step counts across seeds. We will clarify this.
>
> 5.  **Data Augmentation & Hyperparameter Selection:** We acknowledge the limitation of optimizing augmentations only for RAD and applying that strategy universally. This was a trade-off for controlled comparison vs. method-specific tuning. Similarly, hyperparameters were tuned for training efficiency. Evaluating based on generalization is important future work. We will clarify these limitations.
>
> 6.  **Fairness of Pretraining Comparison (PPO):** We agree it's not a direct 'from scratch' comparison. The intent was to assess the impact of leveraging pretrained features (ID/OOD) vs. learning purely from the RL signal in SPGym. We will clarify this motivation.
>
> 7.  **Generalization Evaluation (Table 3 & Pool Size Choice):** The consistent OOD failure across methods and pool sizes is a key diagnostic finding about current end-to-end visual RL limitations. Our rationale for focusing Table 3 on pool size 5, preliminary findings on larger pools, and the interpretation of the OOD results are discussed in detail in our response to **Reviewer ukCk** (due to space limitations).
>
> 8.  **Procedural Generation:** We agree the current results don't show a demonstrated advantage over ImageNet. We will moderate the claim, framing it as a feature with potential benefits (diversity, control, storage) for future investigation.
>
> ## 3. Other Points
>
> *   **Missing References:** Thank you. We will add CARLA and COOM to the related work discussion.
> *   **Code:** Code is in the supplementary material, omitted from the paper for anonymity. We will add links in the camera-ready version.
> *   **Minor Points:** We will refine terminology (model-based definition, visual diversity) and acknowledge unevaluated features (other modalities) and future analysis directions (image difficulty).
>
> We hope these responses and planned revisions address the reviewer's concerns. We value the feedback and are committed to improving the paper. We appreciate the reviewer's openness to reconsidering their evaluation.

---

> > ### Comment · Reviewer_o6ZL · 2025-04-06
> >
> > 1. Since the **Manhattan distance** reward function has been widely used in prior literature, and I don’t see a better alternative, I consider this concern resolved.
> > 2. Since SOTA methods like DreamerV3 already perform close to the optimal solution, there is limited room for further improvement in terms of training performance, particularly in settings with moderate image pools. As a result, future progress in SPGym will likely center on sample efficiency and generalization performance during evaluation, rather than achieving more optimal training outcomes.
> > 3. I agree with reviewer **ukCk**’s suggestion to incorporate levels of evaluation difficulty. The jump from 100% accuracy on ID to 0% on OOD is too abrupt, disabling a meaningful comparison of existing methods. There’s no telling when a method will be created that surpasses 0% OOD accuracy. Until then, comparisons remain uninformative since ID is too easy and OOD is too hard. Intermediate levels of complexity would enhance SPGym’s utility as an evaluation framework.
> > 4. In your response to **ukCk**, there is no mention of the specific image augmentations applied on the training images. More importantly, across all methods, **evaluation performance decreases as the image pool increases**. This is counterintuitive to the proposition: a larger training pool should, in theory, promote more general representations and improve generalization. However, the results suggest otherwise, as agents appear to perform better on the augmented evaluation setting when overfitting to a single image puzzle. This undermines the claim that the agents are learning truly general representations.
> >
> >     A similar phenomenon occurs when increasing the training set size. As you noted in your response to reviewer **ukCk**,
> >
> >     > we found that with such high visual diversity, the RL training became unstable.
> >     >
> >
> >     This instability likely stems from the agent's inability to learn general representations. Instead, it appears to be exhausting its network capacity, effectively memorizing solutions rather than generalizing across different images.
> >
> >     This is likely also why DreamerV3 performs so much better. It is much more sample-efficient and has a larger network, enabling it to learn solutions for a larger number of images individually.
> >
> > 5. In the linear probe experiment, PPO and SAC are trained end-to-end. This means that the reward signal from policy learning has shaped the encoder weights to provide useful encodings for solving the RL puzzle task. Similar to other results, the performance again drops when the training pool increases, indicating a lack of potential for learning generality. Therefore, I don’t see the isolation of visual representation learning evaluation. However, I do think these results are useful to include in the paper to show that the encoder can be repurposed for classification in such manner.
> > 6. I still believe that only using **grayscale + channel shuffle** is a strong injustice to **CURL** and **SPR**, as they are being employed in ways that deviate from their intended design. This goes beyond a simple lack of hyperparameter tuning. These baselines should either be re-evaluated or omitted.
> >
> > I have raised the score because some of my concerns have been addressed, and I believe SPGym is an interesting problem for visual RL. However, the core issues remain. Most importantly, I remain unconvinced of the disentanglement of representation and policy learning.

---

> > > ### Author Response · Authors · 2025-04-09
> > >
> > > Thank you for the continued engagement and detailed feedback. We appreciate the opportunity to clarify our perspective on the remaining points.
> > >
> > > 1. Our primary claim is that SPGym is a valuable tool for *evaluating* the visual representation learning capabilities of RL agents by *isolating visual diversity as the controlled variable*. While training is end-to-end, comparing the performance of an agent across different pool sizes allows for a controlled comparison of how well their representation learning handles visual stress, because all other task aspects are fixed. It's important to distinguish this claim about *evaluation methodology* from claiming that representation and policy learning are perfectly decoupled during training, or that the learned representations are universally general-purpose – claims we do not make. We will ensure this distinction is clear in the revised manuscript.
> > >
> > > 2. We agree with your assessment that future progress will likely focus on sample efficiency and generalization. We also believe SPGym remains valuable for highlighting the *scaling limits* of current methods, including SOTA ones like DreamerV3 (as seen with pool size 100), in terms of efficiently learning representations solely from the RL objective under increasing visual diversity.
> > >
> > > 3. We agree on the value of intermediate evaluation difficulties and thank you and reviewer ukCk for the suggestion. We plan to incorporate the 'Easy Level' results and discuss the tiered approach as important future work.
> > >
> > > 4. To evaluate performance on the proposed 'Easy Level' OOD setting, we took the trained agents and tested them on augmented versions of their training images. Specifically, we applied the same augmentations we considered in the paper (crop, shift, grayscale, inversion, channel shuffle) to the training images, ran evaluations for each augmentation type individually across all 5 seeds for 100 episodes, averaged the success rate for each augmentation, and then reported the average success rate across all augmentation types. We understand the observation that performance on these augmented images tends to decrease as the training pool size increases seems counter-intuitive. One interpretation is that agents achieving better performance (typically on smaller pools) learned SPGym's specific structural invariances better, resulting in representations that are more robust to these simple geometric or color perturbations of the images they were trained on.
> > >
> > > 5. Regarding the linear probe results, we highlight two key findings: First, there is a statistically significant correlation between the quality of the learned representations (measured by the linear probe's accuracy in decoding the underlying state from observations) and the agent's task performance (measured by steps to reach 80% success). Agents with representations that better encode the state learn the task faster. Second, both the probe accuracy and the task performance systematically degrade as the image pool size increases. Since the only variable changing is the visual diversity introduced by the larger pool, this suggests that increased visual diversity directly hinders the agent's ability to learn high-quality, task-relevant representations from the RL objective alone, consequently impairing policy learning performance. We believe including these results empirically supports our claim that SPGym effectively evaluates representation learning under varying visual diversity.
> > >
> > > 6. Thank you for pushing on the augmentation strategy for CURL/SPR. Prompted by your valid concern about fairness, we conducted a dedicated augmentation search specifically for CURL and SPR, following the protocol we describe Appendix C.2. This search included the same augmentations considered for RAD, and we also included shift + color jitter, which was used originally by SPR. Empirically, we found that for this specific task in SPGym, the combination of grayscale + channel shuffle still yielded the best sample efficiency for both methods. Our interpretation remains that preserving tile structure is particularly critical here. We will add this clarification and the supporting evidence to the Appendix. Anonymized links to learning curves: [CURL Augmentation Search](https://i.postimg.cc/Gmj3q5JF/curl-aug-search.png), and [SPR Augmentation Search](https://i.postimg.cc/VNYzCLGy/spr-aug-search.png)
> > >
> > > We hope these clarifications fully address your remaining concerns. We have incorporated much of your feedback into our revision plans and believe the paper, with these changes, makes a valuable contribution. Thank you again for your time and insightful comments.

---

### Decision · Program_Chairs · 2025-05-01

**Decision:**

Accept (poster)

**Comment:**

This paper presents a new RL benchmark along with experiments on existing model-free and model-based RL algorithms. All reviewers appreciate the quality of the contribution, even though a few elements could be improved and clarified. Reviewer o6ZL and ukCk mention the the levels of evaluation difficulty is limited and that it is unclear why the evaluation performance decreases as the image pool increases (it is likely not testing generalization in this paper but the capacity to overfit). Reviewer o6ZL mentions some experimental designs elements that could be improved.